# ChemX: A Collection of Chemistry Datasets for Benchmarking Automated Information Extraction

**Anastasia Vepreva**[1]    **Julia Razlivina**[1]

**Maria Eremeeva**[1]    **Nina Gubina**[1]    **Anastasia Orlova**[1]    **Aleksei Dmitrenko**[1]

**Ksenya Kapranova**[1]    **Susan Jyakhwo**[1]    **Nikita Vasilev**[1]    **Arsen Sarkisyan**[1]

**Ivan Yu. Chernyshov**[1]    **Vladimir Vinogradov**[1]    **Andrei Dmitrenko**[1,2]

[1]Center for AI in Chemistry, ITMO University, St. Petersburg, Russia
[2]D ONE AG, Zurich, Switzerland

dmitrenko@pish.itmo.ru

## Abstract

Despite recent advances in machine learning, many scientific discoveries in chemistry still rely on manually curated datasets extracted from the scientific literature. Automation of information extraction in specialized chemistry domains has the potential to scale up machine learning applications and improve the quality of predictions, enabling data-driven scientific discoveries at a faster pace. In this paper, we present ChemX, a collection of 10 benchmarking datasets across several domains of chemistry providing a reliable basis for evaluating and fine-tuning automated information extraction methods. The datasets encompassing various properties of small molecules and nanomaterials have been manually extracted from peer-reviewed publications and systematically validated by domain experts through a cross-verification procedure allowing for identification and correction of errors at sources. In order to demonstrate the utility of the resulting datasets, we evaluate the extraction performance of the state-of-the-art large language models (LLMs). Moreover, we design our own agentic approach to take full control of the document preprocessing before LLM-based information extraction. Finally, we apply the recently emerged multi-agent systems specialized in chemistry to compare performance against the strong baselines. Our empirical results highlight persistent challenges in chemical information extraction, particularly in handling domain-specific terminology, complex tabular and schematic formats, and context-dependent ambiguities. We discuss the importance of expert data validation, the nuances of the evaluation pipeline, and the prospects of automated information extraction in chemistry. Finally, we provide open documentation including standardized schemas and provenance metadata, as well as the code and other materials to ensure reproducibility. ChemX is poised to advance automatic information extraction in chemistry by challenging the quality and generalization capabilities of existing methods, as well as providing insights into evaluation strategies.

## 1 Introduction

Integration of machine learning (ML) and artificial intelligence (AI) into chemistry has produced a series of revolutionary works in drug discovery, materials science, and molecular modeling. A key driver of this progress is the availability of robust benchmark datasets that provide the essential

foundation for training, evaluating, and refining computational models. By offering standardized metrics for comparison, such datasets help researchers gauge algorithmic performance, uncover limitations in existing methods, and accelerate advancements in the field [1, 2, 3, 4, 5]. Naturally, domain-specific datasets remain relatively low-scale but play an equally important role in advancing more specialized areas of chemistry. For example, researchers have demonstrated the efficacy of ML in predicting cellular toxicity of inorganic nanomaterials [6], forecasting exchange bias in magnetic heterostructures [7], and designing nanozymes with tunable catalytic activity [8]. These studies underscore the critical role of high-quality data gathering and curation pipelines to enable reliable predictive modeling. More importantly, they make a strong case for advancing automated information extraction solutions to scale up data-driven scientific discoveries.

Early efforts in automated data extraction relied on rule-based systems and dictionary matching, which struggled with the linguistic diversity and contextual nuances of scientific texts. For example, tools like OSCAR4 [9] and ChemDataExtractor [10] utilized predefined grammars and regular expressions to identify entities such as chemical compounds, properties, and reaction conditions. However, these approaches encounter difficulties due to the wide range of topics and reporting formats in chemistry and materials research, as they are specifically tailored for narrow use cases.

Chemical and biomedical named entity recognition (NER) tasks have historically relied on corpora like CHEMDNER [11] and ChemProt [12], which focus on token-level classification and relation extraction, respectively. Deep learning architectures such as ChemBERTa and domain-tuned transformers have established baselines in this space [13, 14, 15, 16]. For solid-state materials, research on information extraction has focused on several key areas, including NER of chemical synthesis parameters from methods sections [17, 18, 19], identification of peak absorption wavelengths in UV-Vis experiments [20] and other relevant data extraction tasks [21, 22, 23, 24]. More recently, the CLUB benchmark [25] expanded entity tasks to both patents and literature, though it remains text-only and modality-restricted. These resources, while foundational, lack support for figure-grounded or multimodal entity linking that real-world documents demand.

The recent wave of LLMs and vision-language pretraining offers a promising path forward, enabling systems capable of reasoning over multimodal data [26, 27]. Such works as [28, 29, 30] focus on LLM fine-tuning for structured information extraction from scientific text, including the domain of solid materials. Benchmarks such as AgentBench [31] and MultiAgentBench [32] evaluate autonomous LLM-based agents across interactive and multimodal tasks. The development of multi-agent systems inevitably necessitates the establishment of appropriate benchmarks. Solovev *et al.* [33] propose a multi-agent system designed for drug discovery. To support this work, they construct six datasets containing drug-related properties, thereby underscoring the importance of high-quality and well-validated datasets. Notably, existing benchmarks fail to adequately assess the performance of automated chemical information extraction systems, which represents a critical gap that our work addresses. Current datasets lack the domain-specific rigor and multimodal scope required to evaluate such tasks in specialized chemical subfields. Recent works such as ChemLLM [34] and ORDerly [35] aim to standardize evaluation and data preparation in chemistry while emphasizing multimodal and reproducible frameworks.

To advance automated data extraction in chemistry, we present ChemX, a manually curated multimodal benchmark dataset aimed at extracting chemical features from textual and visual content across diverse chemical domains. By capturing the heterogeneity and interconnectedness of real-world chemical literature, ChemX provides a foundation for evaluating and training models that bridge traditional NLP with vision-language reasoning, large language models, and collaborative multi-agent systems. This work makes two major contributions:

- We provide the ChemX benchmark, a collection of 10 curated datasets describing various properties of nanomaterials and small molecules. Each dataset is accompanied with detailed documentation, standardized metadata, and cross-verification by domain experts. The datasets are available as a HuggingFace collection, and the corresponding documentation can be accessed via https://ai-chem.github.io/ChemX.

- In this work, we also present a systematic evaluation of state-of-the-art LLMs and agentic systems in the task of automated information extraction from domain-specific scientific literature. The code for the extraction experiments is provided in the GitHub repository.

## 2 Related Works

There is a growing ecosystem of benchmark datasets in the chemical sciences, many of which are designed to support machine learning models for property prediction, structural analysis, or vision-language tasks [36, 37, 38, 39, 40, 41]. While these studies have significantly advanced property prediction in chemistry, they are not designed to benchmark the performance of automated information extraction systems.

In the realm of chemical text understanding, CLUB [25] delivers four benchmark datasets for token classification and NER tasks across patents and scientific papers, created by chemists. ChemTEB [42] introduces an embedding benchmark tailored for chemical texts, evaluating models on retrieval and semantic similarity. In computational chemistry, the Cuby framework [43] integrates well-established benchmarks like GMTKN55 and NCIAtlas, providing tooling for large-scale simulation comparison. Huang et al. extended this direction by using generative models to predict inorganic synthesis conditions, demonstrating the potential of deep learning for reasoning over complex chemical inputs [44]. Additionally, the FedChem framework introduced a federated learning benchmark for molecular property prediction, simulating real-world data heterogeneity and privacy constraints [45].

While these prior studies established valuable foundations for information extraction in chemistry, it is important to note that they were all conducted before the widespread adoption of modern LLMs. Our work introduces a modern LLM-based benchmark for information extraction in chemistry. Unlike pre-LLM era works, we systematically evaluate state-of-the-art language models models and agentic frameworks, going beyond prior technological constraints.

The most closely related study to our work was conducted by Odobesku et al., who developed nanoMINER for automated data extraction using a manually curated dataset of enzymatic activity of nanomaterials [46, 47]. While their approach demonstrates the feasibility of structured information extraction, it is limited to a single highly specialized application. The authors confirm the need to create more high-quality chemistry datasets for benchmarking similar solutions and improving their generalization capabilities. In this work, we address this need by introducing a collection of 10 datasets suitable for the task. In our evaluation experiments, we challenge modern LLMs, as well as agentic approaches, with information extraction, and include nanoMINER for comparison.

## 3 ChemX

ChemX is a collection of X manually curated benchmarking datasets for automated information eXtraction across two major domains: nanomaterials and small molecules (Figure 1). It is a multimodal benchmark that supports robust chemical information extraction from heterogeneous data — tables, graphs, unstructured text. Each dataset is accompanied with detailed documentation available at this link.

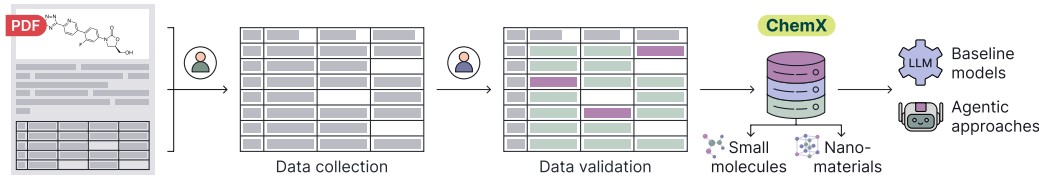

Figure 1: ChemX. This pipeline includes manual collection of multimodal data from scientific articles, further validation by domain experts and benchmarking automated data extraction.

### 3.1 Ontology

An overview of the datasets, including domains, sizes, extracted features, and descriptions, is provided in Table 1. For small molecule datasets, the ontology centers around molecular descriptors, including SMILES representations, biological activity metrics (e.g., MIC, IC$_{50}$), and compound-specific metadata. In contrast, nanomaterials and other material-centric datasets involve a substantially broader set of parameters, encompassing physicochemical properties (e.g., size, zeta potential,

surface coating), synthesis conditions, structural characteristics, and application-specific outcomes. This reflects the inherent complexity and multimodality of material-related information in scientific literature. Including the datasets of varying sizes and complexity in both domains creates a balanced and practical benchmark for automated information extraction. Annotation guidelines and other related details for each dataset are presented on the documentation website.

Table 1: ChemX benchmark datasets grouped by domain.

| Domain | Dataset | Size | Features | | Description |
| | | | String | Numeric | |
| --- | --- | --- | --- | --- | --- |
| Nano-materials | Cytotox | 5535 | 12 | 9 | Cytotoxicity of nanoparticles in normal and cancer cell lines. |
| | Seltox | 3286 | 9 | 14 | Toxic effects of nanoparticles on bacterial strains. |
| | Synergy | 3326 | 10 | 19 | Drug–nanoparticle synergy in antibacterial assays. |
| | Nanozymes | 1135 | 9 | 11 | Catalytic properties of inorganic enzyme mimics. |
| | Nanomag | 2578 | 8 | 16 | Magnetic nanomaterials and their biomedical uses. |
| Small molecules | Benzimidazoles | 1721 | 6 | 1 | SMILES molecules with MICs for antibiotic SAR studies. |
| | Oxazolidinones | 2923 | 6 | 1 | Synthetic antibiotics with biological activity data. |
| | Complexes | 907 | 4 | 1 | Organometallic chelate complexes with thermodynamic parameters. |
| | Eye Drops | 163 | 2 | 1 | Drug permeability data across corneal tissue. |
| | Co-crystals | 70 | 7 | 0 | Drug co-crystals with improved photostability. |

## 3.2 Data Collection

We gathered the data from a broad corpus of peer-reviewed chemistry publications by manual information extraction by domain experts. The information was originally sourced in the text, tables, schematics, drawings, plots, and other types of formats commonly used in the scientific literature Figure 2.

Our domain experts annotated a wide range of targets, encompassing chemical entities (i.e., nanoparticles, organic and inorganic molecules), their synthesis protocols, physicochemical and biomedical properties. Upon data collection, we performed extensive preprocessing of the extracted entities to ensure consistency of machine-readabable formats. For example, chemical structures depicted on figures were manually redrawn using ChemDraw or the PDB Chemical Sketch Tool to ensure accurate conversion to SMILES. Molecular names referenced in the text were also converted to SMILES notation using the PubChem API. As a result, ChemX is built upon over 1,500 annotated articles spanning two chemistry domains, namely, small molecules and nanomaterials.

## 3.3 Quality Control

To evaluate data integrity, we applied a stratified manual cross-verification procedure depicted on Figure 5. From each source article represented in a dataset, approximately 20% of entries were randomly selected and reviewed against the original source material, including PDFs, figures, and supplementary tables. Sampling was rounded up to ensure that at least one entry from each source article was manually reviewed during the verification process. Errors — including transcription mistakes, structural mismatches, unit inconsistencies, and unsupported inferences — were categorized

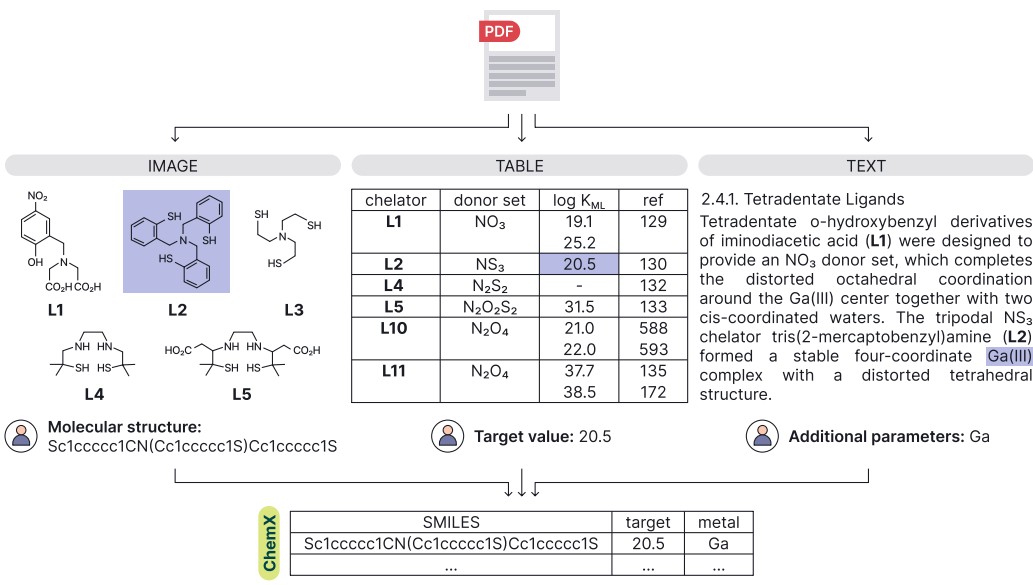

Figure 2: Multimodal data extraction: A real-world example for collecting a dataset with chelate complexes [48]

as either common (recurring patterns) or isolated (single occurrence). Importantly, if an isolated error was identified during review, we systematically checked all the other entries from the same source article, even if they were not part of the original sample. This additional step was intended to determine whether similar issues occurred in other records from the same publication. Error categorization informed the correction strategy. For common errors, we formulated rule-based recommendations that specified the field affected, the observed scope of recurrence, and the appropriate method for correction, such as structural replacement, unit standardization, or removal of inferred content. Corrections were then applied across the whole group. All recommendations were documented in writing and communicated to the dataset curators for implementation across relevant records. Isolated issues were corrected individually.

## 3.4 Dataset Overview and Analysis

The number of openly accessible articles for each dataset is presented in Figure 3B. The distribution of publication years (Figure 3A) reflects the growth of the underlying literature since the early 2000s, with a marked rise in publications over the past decade.

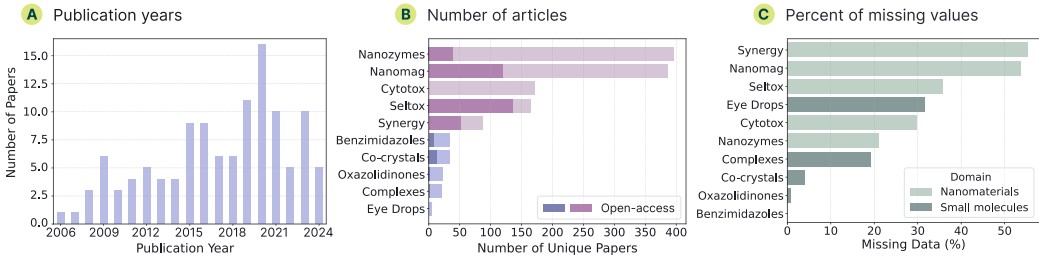

Figure 3: Overview of ChemX. (A) Distribution of the publication years. (B) Number of articles per dataset. (C) Percent of missing values per dataset.

Furthermore, we quantified the prevalence of missing values across the datasets (Figure 3C). Certain datasets exhibit substantial sparsity due to incomplete data reporting in the original publications.

This heterogeneity in data completeness is advantageous for benchmarking purposes, as it facilitates rigorous evaluation of automated extraction systems. Specifically, such variability enables assessment of both the accurate retrieval of reported information and the correct identification of absent data, ensuring robust performance metrics.

### 3.5 Labeling datasets by complexity level for extraction

We assess dataset extraction complexity using five key criteria. A primary challenge is heterogeneous formats, where data is dispersed across text, tables, and complex figures, making parsing difficult. Non-uniform table structures often require cross-referencing with the main text. Semantic ambiguity in labels and units demands contextual interpretation. Furthermore, multi-value records need careful linking of values to their correct materials and units, increasing error risk compared to single-value entries. Finally, domain differences matter; extracting hierarchical relationships for nanomaterials is more complex than using standardized encodings for small molecules.

Table 2: Classification of datasets by complexity level

| Domain | Dataset | Complexity |
|---|---|---|
| Nanomaterials | Cytotox | High |
| | Seltox | High |
| | Synergy | High |
| | Nanomag | High |
| | Nanozymes | Medium |
| Small molecules | Benzimidazoles | Medium |
| | Oxazolidinones | Medium |
| | Co-crystalls | Medium |
| | Eye drops | Low |
| | Complexes | Low |

Datasets are classified as low, medium, or high complexity based on these factors, with multi-format parsing, irregular tables, multi-value linking, and hierarchical relationships elevating the difficulty.

## 4 Experiments

We performed a series of experiments to evaluate the performance of LLMs in extracting structured data from scientific articles. The study compared two distinct approaches: (1) LLMs as baseline models and (2) agentic approaches. To quantitatively assess the quality of extraction, we calculated the precision, recall, and F1-score for each extracted parameter. For this, we calculated the following:

- **True Positives (TP):** The count of values correctly extracted (i.e., the value exists in both the original dataset and the extracted dataset).

- **False Positives (FP):** The count of values incorrectly extracted (i.e., the value does not exist in the original dataset but is present in the extracted dataset).

- **False Negatives (FN):** The count of missing values (i.e., the value exists in the original dataset but is absent from the extracted dataset).

For each PDF in the dataset, we computed precision, recall, and F1 score based on those quantities. The resulting metrics were then aggregated across all PDFs in the dataset and averaged by dividing the total sum by the number of PDFs.

To standardize inputs, we created the following prompt template:

**system_prompt** = "You are a domain-specific chemical information extraction assistant. You specialize in the chemistry of ... . Your area of expertise includes ... ."
**user_prompt** = "Your task is to extract **every** mention of ... for ... from a scientific article, and output a **JSON array** of objects **only** (no markdown, no commentary, no extra text):

1. *Feature 1 (string)*: *Description* (e.g., *'example'*).

2. *Feature 2 (numeric)*: *Description* (e.g., *'example'*).

3. ...

4. *Target value (numeric)*: *Description* (e.g., *'example'*).

Extraction rules:

- Extract **each** ... mention as a separate object.

- Do **not** filter, group, summarize, or deduplicate. Include repeated mentions and duplicates if they occur in different contexts.

- If you cannot find a required field for an object, re-check the context; if it's still absent, set that field's value to "NOT_DETECTED"

- *Other rules specific to this dataset*

- The example of JSON below shows only one extracted samples, however your output should contain **all** mentions of ... for ... present in the article.

Output **must** be a single JSON array, like: [{ "feature 1": "example of feature 1", "feature 2": "example of feature 2", ... "target value": "example of target value" }]"

Specific prompts for each dataset can be found in the section 9.5.

## 4.1 Baseline models

GPT-4o was selected due to the advanced multimodal data processing capabilities. Experiments were conducted using exclusively open-access scholarly articles to ensure reproducibility and compliance with accessibility standards. Articles were processed either as full-text PDF files or as sets of JPEG images, with extraction performance metrics computed independently for each input modality. In cases where supplementary materials contained relevant information, the primary article and supplementary files were merged into a single composite document prior to processing. The Assistants API GPT-4o was leveraged to enhance reproducibility across extraction workflows. [49]

## 4.2 Agentic approaches

### 4.2.1 Single agent

To address the opacity and inconsistency of OpenAI's black-box PDF and screenshot processing, we adopt a single-agent preprocessing approach using the marker-pdf SDK [50]. The marker-pdf library was selected due to its robust capabilities for accurately preserving document structure and semantic integrity during extraction. The text and tables are converted into markdown format, while local image paths are generated for images and inserted into their corresponding positions within the markdown document.

Each extracted image is then processed by the `gpt-4o-2024-11-20` model using a tailored image description prompt. GPT-4o's strong multimodal capabilities enable accurate interpretation of diverse image types, ensuring consistent descriptions. In addition, this design choice made it possible to fairly compare the single agent approach with baseline models, isolating the factor of document pre-processing. These are inserted into the markdown within `<DESCRIPTION_FROM_IMAGE>` tags, producing a `described.md` file. Finally, the described markdown is processed by `gpt-4.1-mini-2025-04-14` model for information extraction. This pipeline allows for a controlled, semantically faithful preprocessing workflow and fair comparison against baseline models. The final outputs are compiled into dataset-specific CSV files.

#### 4.2.2 The multi-agent approach

We included nanoMINER for comparison to benchmark its performance against the other approaches. Notably, nanoMINER is a highly specialized solution in the nanomaterial domain that consists of three agents, two of which were fine-tuned to extract properties of nanozymes. More specifically, the vision agent leveraged the fine-tuned YOLO model to recognize plots of enzymatic parameters, while the NER agent made use of the fine-tuned Llama and Mistral models to better recognize nanozyme properties in text [46]. Therefore, while providing unmatched performance for the nanozymes dataset, nanoMINER could not be easily evaluated on the other datasets of ChemX.

## 5 Experimental results

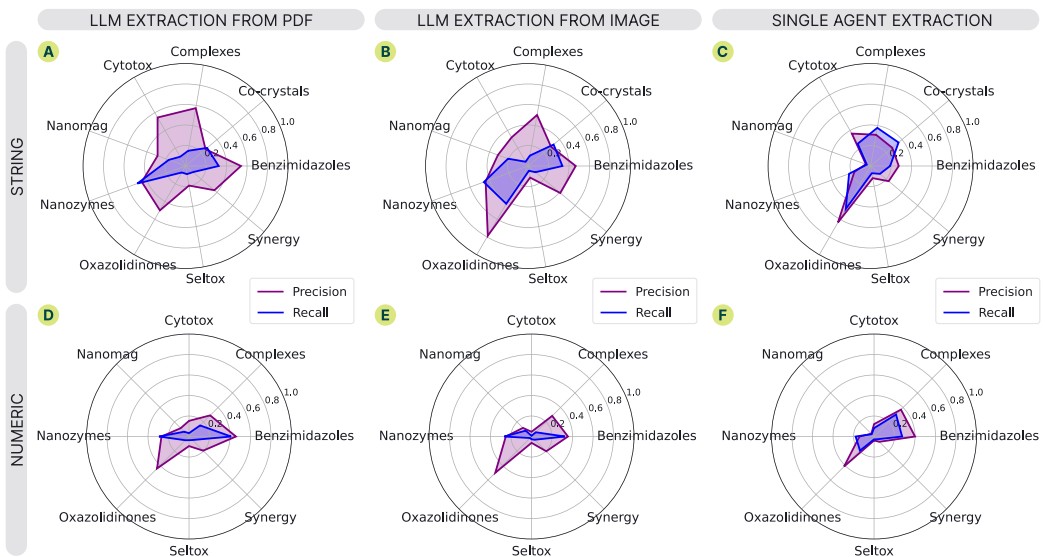

Figure 4: Extraction metrics for LLM and single agent approach. For LLM method two distinct data extraction approaches were evaluated: text parsing from PDF files versus image processing from JPEG files. (A-C) Recall and precision for string values extraction using LLM extraction from PDF, LLM extraction from images and single agent extraction, respectively. (D-F) Recall and precision for numeric values extraction using LLM extraction from PDF, LLM extraction from images and single agent extraction, respectively. The eye drops dataset was excluded from the analysis because it does not include open-access articles. The co-crystals dataset was omitted when calculating metrics for numerical parameters, as it lacks numerical values. Average metrics across all columns in datasets are presented. Standard deviation is not presented for better display.

For the baseline models, we compared two extraction methodologies: providing LLMs with PDF files versus JPEG images for processing. In addition to the baseline, we included the single agent approach for comparison (Figure 4). The eye drops dataset was excluded from our analysis due to the absence of open-access articles. Our empirical results evaluating automated information extraction from the nine remaining datasets highlight three key findings:

1. All methods exhibited better performance on textual parameters (such as compound identifiers, compound names, but not SMILES) compared to numerical ones (such as concentration values, etc.).

2. While GPT-4o achieved higher metrics than our single agent suggesting their opaque PDF preprocessing works better, its overall performance remains unsatisfactory for practical applications. Notably, single agent does demonstrate higher recall and F1 values on some datasets (see section 9.4), which prompts further investigation of the factors impacting extraction performance.

3. We observed consistently higher accuracy for small molecule datasets relative to nanomaterials, which was expected due to a larger number of parameters involved.

We also compared the baseline LLMs and single agent approach with nanoMINER [46], a recently developed multi-agent system for nanomaterials datasets (Table 3). While nanoMINER by far outperforms all baselines in terms of accuracy and F1 score, it remains a highly specialized solution that cannot be directly applied to the other datasets of the same domain. LLMs show better metrics compared to the single agent approach for all datasets but Cytotox. However, both approaches remain impractical due to insufficient extraction quality.

Table 3: Comparison of different approaches in the nanomaterial domain of ChemX. For LLM-based extraction, the best metrics between PDF and JPEG are presented. The metrics for nanoMINER are taken from the original paper. Mean value and standard deviation for precision and F1-score are presented; these values do not account for the stochastic variability inherent to LLM outputs. The observed high extraction error rates primarily stem from the inherent heterogeneity in data quality across the numerous columns present in nanomaterial datasets.

| | LLM | | Single agent | | nanoMINER | |
|---|---|---|---|---|---|---|
| | Precision | F1-score | Precision | F1-score | Precision | F1-score |
| Nanozymes | $0.35 \pm 0.21$ | $0.34 \pm 0.20$ | $0.15 \pm 0.11$ | $0.16 \pm 0.12$ | $0.80 \pm 0.16$ | $0.71 \pm 0.16$ |
| Cytotox | $0.38 \pm 0.27$ | $0.11 \pm 0.07$ | $0.26 \pm 0.19$ | $0.18 \pm 0.13$ | - | - |
| Seltox | $0.13 \pm 0.08$ | $0.07 \pm 0.04$ | $0.07 \pm 0.07$ | $0.05 \pm 0.04$ | - | - |
| Synergy | $0.27 \pm 0.24$ | $0.09 \pm 0.08$ | $0.14 \pm 0.17$ | $0.08 \pm 0.10$ | - | - |
| Nanomag | $0.18 \pm 0.18$ | $0.13 \pm 0.13$ | $0.04 \pm 0.05$ | $0.03 \pm 0.03$ | - | - |

## 6 Discussion

We introduce ChemX, a curated collection of 10 benchmarking datasets spanning small molecules and nanomaterials, rigorously validated through expert cross-verification procedure to ensure reliable evaluation of information extraction methods. Our analysis demonstrates its utility through evaluations of modern LLMs, a custom single-agent pipeline, as well as the state-of-the-art multi-agent system for the nanomaterials domain. Our findings reveal a variety of persistent challenges discussed below.

### 6.1 Cross-verification results

The primary objective of dataset validation is to verify that the data is suitable for automated extraction tasks by identifying and correcting unreasonable or erroneous data values. To ensure comprehensive quality control, we employed a stratified sampling strategy, wherein each article was reviewed at least once. Errors were categorized into recurring and isolated types, with a focus on addressing common issues, as these constituted the majority of required corrections. A key strength of this validation procedure is its ability to extrapolate correction rules across the entire dataset based on a limited subset of manually verified examples. The limitations of this procedure include execution of only a single testing cycle. We provide supplementary figures summarizing correction statistics for all ten datasets (available in subsection 9.2). Across all datasets, the proportion of corrected values remained below 4%, confirming overall data reliability. Therefore, most of the data was initially accurate and the validation procedure effectively eliminated remaining deviations.

### 6.2 Extraction quality assessment

To evaluate extracted string values, we combined semantic and character-level similarity measures. For semantic assessment, we used SentenceTransformer (`BAAI/bge-base-en-v1.5` on Hugging-Face) to compute cosine similarity between embeddings, supplemented by Levenshtein distance calculations via RapidFuzz for character-level comparison. However, establishing accurate string alignments proved challenging, as both methods frequently produced false matches, particularly for numerical data, chemical formulas, and units where semantic relationships are poorly captured. This limitation highlights the need for specialized matching strategies in chemical information extraction.

The comparative analysis reveals consistently lower accuracy for numerical parameters and strong heterogeneity across columns (Table 4 and Tables 5–9 in Appendix 9.4). In molecular datasets, SMILES strings showed zero accuracy, whereas compound identifiers and target types were extracted almost perfectly. Nanomaterial datasets exhibited greater variance due to complex tabular formats and inconsistent terminology. This heterogeneity likely stems from OCR errors, unit ambiguities, and strict precision requirements for numerical values. Additional factors in nanomaterial datasets—heterogeneous reporting, non-standardized terms, and frequent missing values—further reduce accuracy and increase false negatives. Overall, aggregated F1 metrics obscure these disparities; future evaluations should therefore include both aggregate and column-level results.

## 6.3 Current methodological limitations

Recent studies, such as ChemCrow [51] and FutureHouse [52], have demonstrated the potential of LLMs for automated data extraction, though this is not their primary objective. In a preliminary test, FutureHouse was applied to a single article containing one sample from the nanozyme dataset (Table 4). After processing for 3 minutes and 42 seconds, it produced a reasonably accurate output (Case 1, Table 4). However, when subsequently provided with another article containing two nanozyme samples as a follow-up task, the system encountered significant failures. The extraction process took nearly 16 minutes and returned mostly null (NaN) values. In constrast, nanoMINER excels in these tasks, but remains limited to a single application, as mentioned earlier.

Our extraction experiments underscore the inherent constraints of general-purpose LLMs in chemical structure recognition (Section 9.4). Although specialized tools like DECIMER [53] can convert molecular images to SMILES strings, their practical integration remains unfeasible due to two unresolved technical challenges: (1) reliable detection of discrete molecular depictions within complex article layouts, and (2) accurate segmentation of heterogeneous image formats. Future developments in computer vision—particularly for automated molecule localization and standardized image preprocessing—may eventually enable DECIMER's incorporation into extraction pipelines. However, given these current limitations, we deliberately excluded such tools from our experiments.

## 6.4 Prospects of automated information extraction

Our findings demonstrate that, despite recent advances in AI and agentic systems, the accurate extraction of chemical information remains a surprisingly complex task that requires significant innovation to be effectively addressed. ChemX has already been utilized to benchmark agent-based automated data extraction systems; however, the performance results were suboptimal, further highlighting the challenges inherent in this domain [54]. On one hand, future research should focus on the generalization capabilities of highly specialized systems, such as nanoMINER, to enable their seamless application to other datasets within the same domain. On the other hand, agents in such systems should be equipped with more specialized tools, such as DECIMER, to effectively handle real-world applications. In any case, the future of automated information extraction appears to be multi-agent, and greater efforts from the research community should be directed toward agent orchestration.

## 7 Conclusion

ChemX is a curated benchmark comprising 10 rigorously validated datasets encompassing small molecules and nanomaterials. Each dataset was cross-verified by domain experts to ensure robustness in the assessment of information extraction methodologies. We demonstrated the utility of ChemX through a series of evaluations, including state-of-the-art LLMs, a custom single-agent approach, and the recently proposed multi-agent system specialized on nanomaterials. We showed that modern LLMs and agent-based approaches exhibit significant limitations in performing extraction tasks on ChemX. Analyzing the experimental results, we identified key challenges inherent to chemical and nanomaterial data—including heterogeneous representations, lack of standardized nanomaterial descriptions, and prevalent missing values—hindering robust automated information extraction performance. As the first benchmarking resource of its kind, ChemX provides a critical foundation for advancing automated information extraction in chemistry. By offering rigorously validated, expert-curated datasets, it enables systematic evaluation and refinement of emerging techniques, ultimately driving the progress in chemical information extraction.

# 8 Acknowledgment

This work supported by the Ministry of Economic Development of the Russian Federation (IGK 000000C313925P4C0002), agreement No139-15-2025-010.

We sincerely thank Olga Kononova for constructive feedback and fruitful discussions that helped us improve the manuscript.

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

# 9 Appendix

## 9.1 Validation of datasets by domain experts

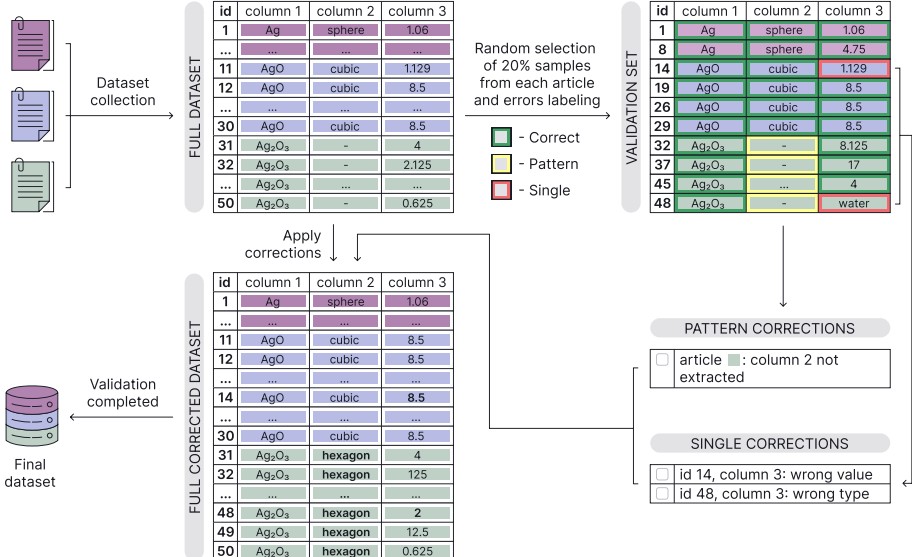

Figure 5: Quality control process for ChemX datasets

## 9.2 Error statistics

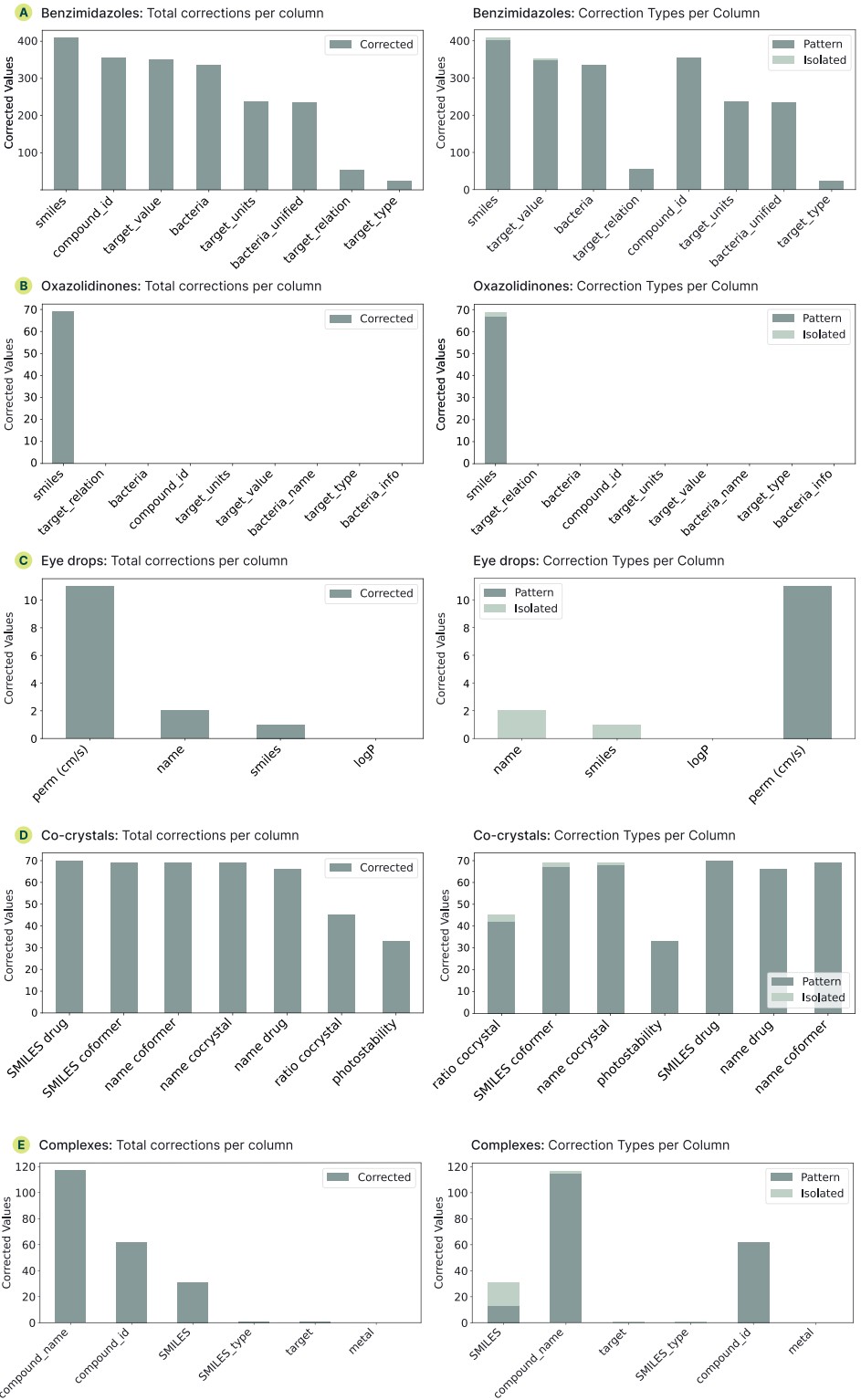

Figure 6: Error statistics for small molecules datasets

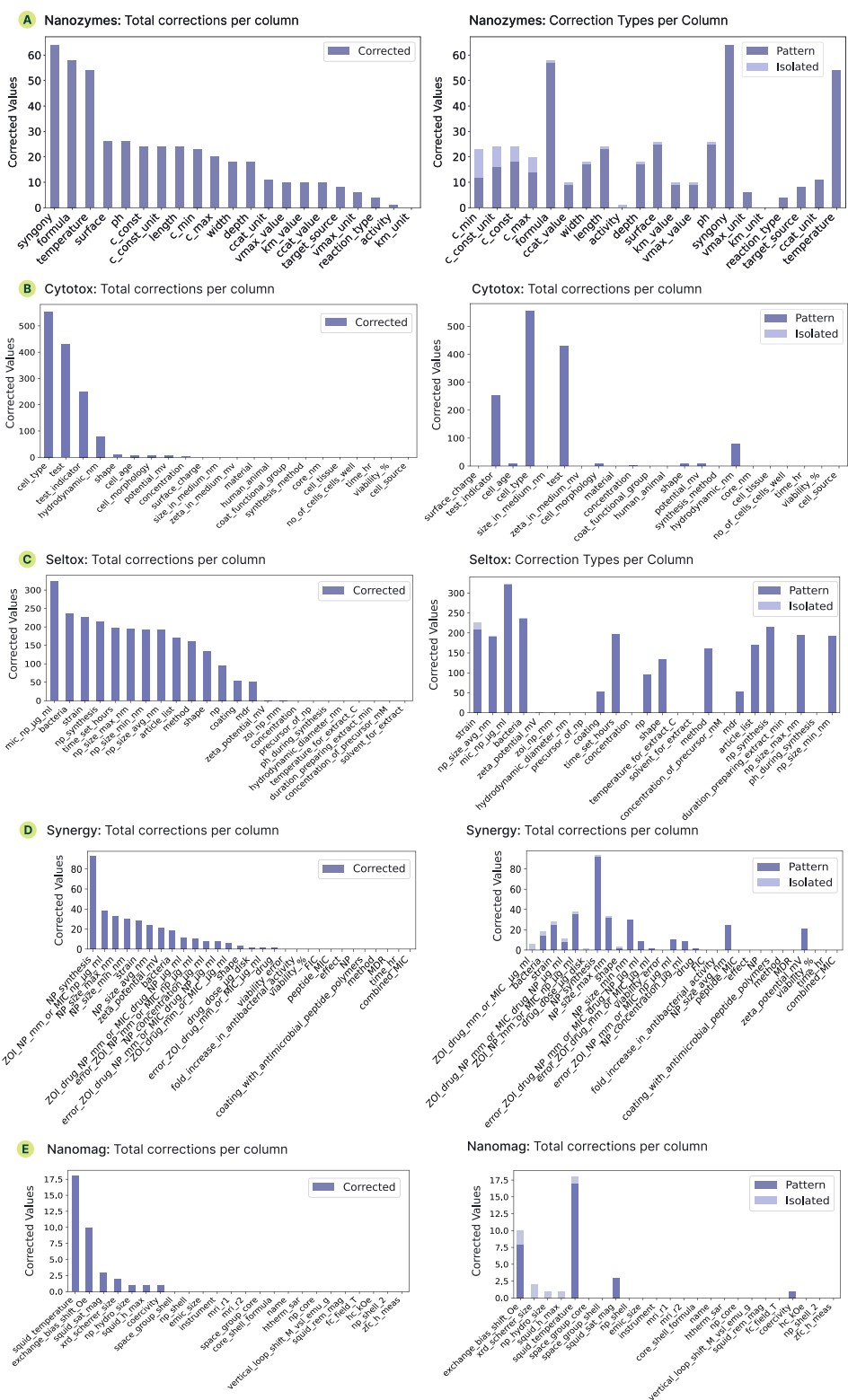

Figure 7: Error statistics for nanomaterials datasets

## 9.3 FutureHouse

Table 4: Extraction results for the nanozymes dataset using the Crow agent of the FutureHouse platform. Case 1: article, type - task, duration - 3m 42s. Case 2: article, type - follow up, duration - 15m 42s.

| Case | Extracted | Expected |
|---|---|---|
| 1 | ```json
[
{
"Formula": "MnO2 (1.1)",
"Activity Type": "Oxidase (1.1)",
"Crystal System": "NaN (1.1, 1.2)",
"Length (nm)": "NaN (1.1)",
"Width (nm)": "NaN (1.1)",
"Depth (nm)": "NaN (1.1)",
"Surface Molecule": "NaN (1.1, 1.2)",
"Km Value": 0.005,
"Km Unit": "mM (1.2)",
"Vmax Value": 0.155,
"Vmax Unit": "U/mL (1.2)",
"Reaction Type":
"2,6-DMP + O2 (oxidase-like oxidation
of 2,6-DMP without H2O2) (1.1, 1.2)",
"C_min (mM)": 0.005,
"C_max (mM)": 1.0,
"Co-substrate Concentration Value":
"NaN (dissolved oxygen used as electron acceptor;
no explicit concentration provided) (1.1)",
"Co-substrate Unit": "NaN (1.1)",
"Catalyst Concentration Value": 0.1,
"Catalyst Unit": "mg/mL (1.3)",
"pH Level": 3.8,
"Temperature (°C)": 25
}
]
``` | ```json
[
{
"Formula": "MnO2",
"Activity Type": "Oxidase",
"Crystal System": "NOT_DETECTED",
"Length (nm)": "NOT_DETECTED",
"Width (nm)": "NOT_DETECTED",
"Depth (nm)": "NOT_DETECTED",
"Surface Molecule": "NOT_DETECTED",
"Km Value": 0.005,
"Km Unit": "NOT_DETECTED",
"Vmax Value": 0.155,
"Vmax Unit": "NOT_DETECTED",
"Reaction Type": "2,6-DMP",
"C_min (mM)": 0.005,
"C_max (mM)": 0.2,
"Co-substrate Concentration Value":
"NOT_DETECTED",
"Co-substrate Unit": "NOT_DETECTED",
"Catalyst Concentration Value": 0.1,
"Catalyst Unit": "mg mL-1",
"pH Level": 3.8,
"Temperature (°C)": 25
}
]
``` |

```json
[
  {
    "Formula": "MnO2",
    "Activity Type": "Oxidase-like",
    "Crystal System": "NaN",
    "Length (nm)": "NaN",
    "Width (nm)": "NaN",
    "Depth (nm)": "NaN",
    "Surface Molecule": "NaN",
    "Km Value": 0.005,
    "Km Unit": "mM",
    "Vmax Value": 0.155,
    "Vmax Unit": "U/mL",
    "Reaction Type": "2,6-DMP + O2",
    "C_min (mM)": 0.005,
    "C_max (mM)": 0.2,
    "Co-substrate Concentration Value": "NaN",
    "Co-substrate Unit": "NaN",
    "Catalyst Concentration Value": 0.1,
    "Catalyst Unit": "mg/mL",
    "pH Level": 3.8,
    "Temperature (°C)": 25,
    "Citation": "1.1, 1.2"
  },
  {
    "Formula": "Au–Ag–Pt",
    "Activity Type": "Peroxidase-like",
    "Crystal System": "NaN",
    "Length (nm)": "NaN",
    "Width (nm)": "NaN",
    "Depth (nm)": "NaN",
    "Surface Molecule": "chitosan",
    "Km Value": "NaN",
    "Km Unit": "NaN",
    "Vmax Value": "NaN",
    "Vmax Unit": "NaN",
    "Reaction Type": "TMB + H2O2",
    "C_min (mM)": "NaN",
    "C_max (mM)": "NaN",
    "Co-substrate Concentration Value": "NaN",
    "Co-substrate Unit": "NaN",
    "Catalyst Concentration Value": "NaN",
    "Catalyst Unit": "NaN",
    "pH Level": "NaN",
    "Temperature (°C)": 37,
    "Citation": "2.1 pages 1-2, 2.1 pages 8-9"
  }
]
```

```json
[
  {
    "Formula": "Pt",
    "Activity Type": "peroxidase",
    "Crystal System": "NOT_DETECTED",
    "Length (nm)": 3.21,
    "Width (nm)": 3.21,
    "Depth (nm)": 3.21,
    "Surface Molecule": "Polyethyleneimine (PEI)",
    "Km Value": 2.02,
    "Km Unit": "mM",
    "Vmax Value": 0.115,
    "Vmax Unit": "10-8 M s-1",
    "Reaction Type": "TMB + H2O2",
    "C_min (mM)": 0.01,
    "C_max (mM)": 0.357,
    "Co-substrate Concentration Value": 3,
    "Co-substrate Unit": "M",
    "Catalyst Concentration Value": 6,
    "Catalyst Unit": "µM",
    "pH Level": 4,
    "Temperature (°C)": 30,
  },
  {
    "Formula": "Pt",
    "Activity Type": "peroxidase",
    "Crystal System": "NOT_DETECTED",
    "Length (nm)": 3.21,
    "Width (nm)": 3.21,
    "Depth (nm)": 3.21,
    "Surface Molecule": "Polyethyleneimine (PEI)",
    "Km Value": 43.6,
    "Km Unit": "mM",
    "Vmax Value": 8.5,
    "Vmax Unit": "10-8 M s-1",
    "Reaction Type": "H2O2 + TMB",
    "C_min (mM)": 0.03,
    "C_max (mM)": 0.177,
    "Co-substrate Concentration Value": 0.8,
    "Co-substrate Unit": "mM",
    "Catalyst Concentration Value": 6,
    "Catalyst Unit": "µM",
    "pH Level": 4,
    "Temperature (°C)": 30,
  }
]
```

## 9.4 Extraction metrics

Table 5: Extraction metrics for all columns of small molecule datasets. Baseline model and single agent methods are presented.

| Dataset | Columns | LLM FROM PDF | | LLM FROM JPEG | | SINGLE AGENT | |
|---|---|---|---|---|---|---|---|
| | | Precision | F1 | Precision | F1 | Precision | F1 |
| oxazolidinone | compound_id | 1.00 | 0.25 | 1.00 | 0.54 | 0.97 | **0.81** |
| | smiles | 0.00 | 0.00 | 0.00 | 0.00 | 0.00 | 0.00 |
| | target_type | 1.00 | 0.25 | 1.00 | 0.54 | 0.93 | **0.78** |
| | target_relation | 1.00 | 0.25 | 0.88 | 0.50 | 0.97 | **0.81** |
| | target_value | 0.44 | 0.09 | 0.50 | 0.04 | 0.41 | **0.26** |
| | target_units | 0.00 | 0.00 | 1.00 | 0.54 | 0.00 | 0.00 |
| | bacteria | 0.00 | 0.00 | 0.83 | 0.53 | 0.94 | **0.78** |
| | | Precision | F1 | Precision | F1 | Precision | F1 |
| benzimidazole | compound_id | 0.88 | **0.58** | 0.74 | 0.55 | 0.44 | 0.32 |
| | smiles | 0.00 | 0.00 | 0.03 | 0.00 | 0.00 | 0.00 |
| | target_type | 0.98 | 0.63 | 0.93 | **0.65** | 0.55 | 0.42 |
| | target_relation | 0.98 | 0.63 | 0.84 | **0.64** | 0.55 | 0.42 |
| | target_value | 0.46 | **0.42** | 0.36 | 0.30 | 0.40 | 0.29 |
| | target_units | 0.11 | 0.03 | 0.00 | 0.00 | 0.10 | 0.07 |
| | bacteria | 0.30 | **0.24** | 0.25 | 0.19 | 0.00 | 0.00 |
| | | Precision | F1 | Precision | F1 | Precision | F1 |
| cocrystals | name_cocrystal | 0.70 | **0.69** | 0.66 | 0.69 | 0.64 | 0.68 |
| | ratio_cocrystal | 0.52 | **0.54** | 0.47 | 0.49 | 0.48 | 0.52 |
| | name_drug | 0.35 | 0.34 | 0.42 | 0.44 | 0.46 | **0.49** |
| | SMILES_drug | 0.00 | 0.00 | 0.00 | 0.00 | 0.00 | 0.00 |
| | name_coformer | 0.08 | 0.08 | 0.15 | 0.15 | 0.19 | **0.21** |
| | SMILES_coformer | 0.12 | 0.12 | 0.31 | **0.33** | 0.09 | 0.10 |
| | photostability_change | 0.03 | 0.03 | 0.03 | 0.03 | 0.06 | 0.08 |
| | | Precision | F1 | Precision | F1 | Precision | F1 |
| complexes | compound_id | 1.00 | **0.39** | 0.67 | 0.23 | 0.43 | 0.36 |
| | compound_name | 0.63 | **0.31** | 0.35 | 0.13 | 0.18 | 0.19 |
| | SMILES | 0.00 | 0.00 | 0.02 | 0.01 | 0.00 | 0.00 |
| | SMILES_type | 0.67 | 0.11 | 0.98 | 0.32 | 0.62 | **0.61** |
| | target | 0.29 | 0.19 | 0.28 | 0.10 | 0.37 | 0.29 |

Table 6: Extraction metrics for all columns of nanozymes dataset. Baseline model and single agent methods are presented.

| Dataset | Columns | LLM FROM PDF | | LLM FROM JPEG | | SINGLE AGENT | |
|---|---|---|---|---|---|---|---|
| | | Precision | F1 | Precision | F1 | Precision | F1 |
| | formula | 0.53 | 0.51 | 0.58 | **0.56** | 0.30 | 0.33 |
| | activity | 0.74 | 0.72 | 0.82 | **0.77** | 0.35 | 0.38 |
| | syngony | 0.59 | **0.59** | 0.56 | 0.55 | 0.07 | 0.07 |
| | length | 0.10 | 0.09 | 0.11 | 0.10 | 0.23 | **0.25** |
| | width | 0.10 | 0.09 | 0.09 | 0.08 | 0.19 | **0.19** |
| | depth | 0.10 | 0.09 | 0.06 | 0.05 | 0.00 | 0.00 |
| nanozymes | surface | 0.33 | **0.33** | 0.26 | 0.26 | 0.01 | 0.02 |
| | km_value | 0.52 | **0.52** | 0.51 | 0.49 | 0.27 | 0.29 |
| | km_unit | 0.64 | **0.63** | 0.65 | 0.62 | 0.25 | 0.27 |
| | vmax_value | 0.34 | 0.33 | 0.34 | **0.34** | 0.21 | 0.22 |
| | vmax_unit | 0.29 | **0.29** | 0.18 | 0.16 | 0.14 | 0.14 |
| | reaction_type | 0.57 | **0.55** | 0.47 | 0.44 | 0.25 | 0.28 |
| | c_min | 0.18 | 0.20 | 0.20 | 0.19 | 0.09 | 0.10 |
| | c_max | 0.12 | 0.11 | 0.21 | 0.20 | 0.12 | 0.11 |
| | c_const | 0.22 | 0.21 | 0.26 | **0.25** | 0.13 | 0.12 |
| | c_const_unit | 0.29 | 0.28 | 0.33 | **0.30** | 0.13 | 0.13 |
| | ccat_value | 0.34 | **0.31** | 0.29 | 0.29 | 0.10 | 0.10 |
| | ccat_unit | 0.10 | 0.10 | 0.12 | 0.12 | 0.02 | 0.02 |
| | ph | 0.56 | 0.54 | 0.70 | **0.66** | 0.23 | 0.26 |
| | temperature | 0.39 | **0.37** | 0.00 | 0.00 | 0.00 | 0.00 |

Table 7: Extraction metrics for all columns of cytotoxicity dataset. Baseline model and single agent methods are presented.

| Dataset | Columns | LLM FROM PDF | | LLM FROM JPEG | | SINGLE AGENT | |
|---|---|---|---|---|---|---|---|
| | | Precision | F1 | Precision | F1 | Precision | F1 |
| | material | 0.46 | 0.11 | 0.19 | 0.04 | 0.50 | **0.33** |
| | shape | 0.54 | 0.16 | 0.41 | 0.10 | 0.29 | **0.22** |
| | coat_functional_group | 0.72 | **0.19** | 0.18 | 0.04 | 0.16 | 0.10 |
| | synthesis_method | 0.29 | 0.08 | 0.14 | 0.03 | 0.21 | **0.16** |
| | surface_charge | 0.42 | 0.13 | 0.33 | 0.07 | 0.33 | **0.24** |
| | core_nm | 0.00 | 0.00 | 0.00 | 0.00 | 0.00 | 0.00 |
| | size_in_medium_nm | 0.06 | 0.02 | 0.00 | 0.00 | 0.09 | 0.06 |
| | hydrodynamic_nm | 0.22 | 0.08 | 0.04 | 0.01 | 0.07 | 0.04 |
| | potential_mv | 0.26 | 0.07 | 0.00 | 0.00 | 0.19 | **0.16** |
| cytotoxicity | zeta_in_medium_mv | 0.08 | 0.03 | 0.00 | 0.00 | 0.10 | 0.07 |
| | no_of_cells_cells_well | 0.20 | 0.06 | 0.07 | 0.02 | 0.10 | 0.04 |
| | human_animal | 0.92 | 0.25 | 0.59 | 0.14 | 0.69 | **0.46** |
| | cell_source | 0.86 | 0.24 | 0.52 | 0.12 | 0.64 | **0.44** |
| | cell_tissue | 0.44 | 0.13 | 0.17 | 0.04 | 0.37 | **0.28** |
| | cell_morphology | 0.46 | 0.15 | 0.28 | 0.06 | 0.25 | **0.18** |
| | cell_age | 0.44 | 0.15 | 0.41 | 0.09 | 0.28 | **0.19** |
| | time_hr | 0.00 | 0.00 | 0.00 | 0.00 | 0.38 | **0.28** |
| | concentration | 0.49 | **0.14** | 0.29 | 0.06 | 0.13 | 0.07 |
| | test | 0.64 | 0.20 | 0.28 | 0.07 | 0.42 | **0.28** |
| | test_indicator | 0.35 | 0.12 | 0.35 | 0.09 | 0.24 | **0.18** |
| | viability_% | 0.07 | 0.03 | 0.00 | 0.00 | 0.02 | 0.02 |

Table 8: Extraction metrics for all columns of seltox dataset. Baseline model and single agent methods are presented.

| Dataset | Columns | LLM FROM PDF | | LLM FROM JPEG | | SINGLE AGENT | |
|---|---|---|---|---|---|---|---|
| | | Precision | F1 | Precision | F1 | Precision | F1 |
| | np | 0.24 | **0.13** | 0.12 | 0.06 | 0.21 | 0.12 |
| | coating | 0.24 | **0.10** | 0.12 | 0.05 | 0.12 | 0.08 |
| | bacteria | 0.28 | **0.14** | 0.17 | 0.09 | 0.20 | 0.12 |
| | mdr | 0.21 | **0.10** | 0.11 | 0.06 | 0.15 | 0.08 |
| | strain | 0.20 | **0.10** | 0.13 | 0.07 | 0.12 | 0.07 |
| | np_synthesis | 0.04 | 0.03 | 0.01 | 0.01 | 0.00 | 0.00 |
| | method | 0.26 | **0.14** | 0.15 | 0.08 | 0.19 | 0.11 |
| | mic_np_µg_ml | 0.09 | 0.04 | 0.06 | 0.03 | 0.12 | 0.07 |
| | concentration | 0.07 | 0.03 | 0.04 | 0.01 | 0.00 | 0.00 |
| | zoi_np_mm | 0.10 | 0.04 | 0.05 | 0.02 | 0.03 | 0.02 |
| seltox | np_size_min_nm | 0.18 | 0.09 | 0.06 | 0.02 | 0.04 | 0.03 |
| | np_size_max_nm | 0.16 | 0.09 | 0.07 | 0.03 | 0.05 | 0.03 |
| | np_size_avg_nm | 0.16 | 0.07 | 0.09 | 0.04 | 0.08 | 0.06 |
| | shape | 0.17 | **0.11** | 0.15 | 0.07 | 0.11 | 0.08 |
| | time_set_hours | 0.16 | 0.08 | 0.15 | 0.08 | 0.00 | 0.00 |
| | zeta_potential_mV | 0.04 | 0.03 | 0.03 | 0.02 | 0.03 | 0.03 |
| | solvent_for_extract | 0.14 | 0.06 | 0.07 | 0.03 | 0.02 | 0.01 |
| | temperature_for_extract_C | 0.03 | 0.02 | 0.08 | 0.05 | 0.00 | 0.00 |
| | duration_preparing_extract_min | 0.01 | 0.01 | 0.02 | 0.02 | 0.00 | 0.00 |
| | precursor_of_np | 0.17 | 0.09 | 0.10 | 0.05 | 0.11 | 0.07 |
| | concentration_of_precursor_mM | 0.08 | 0.04 | 0.07 | 0.04 | 0.01 | 0.02 |
| | hydrodynamic_diameter_nm | 0.01 | 0.01 | 0.02 | 0.02 | 0.03 | 0.03 |
| | ph_during_synthesis | 0.01 | 0.00 | 0.00 | 0.00 | 0.00 | 0.00 |

Table 9: Extraction metrics for all columns of synergy dataset. Baseline model and single agent methods are presented.

| Dataset | Columns | LLM FROM PDF | | LLM FROM JPEG | | SINGLE AGENT | |
|---|---|---|---|---|---|---|---|
| | | Precision | F1 | Precision | F1 | Precision | F1 |
| | NP | 0.46 | 0.11 | 0.57 | **0.19** | 0.24 | 0.14 |
| | bacteria | 0.00 | 0.00 | 0.00 | 0.00 | 0.64 | **0.39** |
| | strain | 0.55 | **0.16** | 0.51 | 0.15 | 0.15 | 0.08 |
| | NP_synthesis | 0.00 | 0.00 | 0.02 | 0.00 | 0.02 | 0.01 |
| | drug | 0.73 | 0.26 | 0.72 | 0.25 | 0.46 | **0.28** |
| | drug_dose_µg_disk | 0.34 | 0.12 | 0.36 | **0.13** | 0.04 | 0.01 |
| | NP_concentration_µg_ml | 0.32 | 0.10 | 0.36 | **0.12** | 0.03 | 0.02 |
| | NP_size_min_nm | 0.04 | 0.01 | 0.01 | 0.00 | 0.35 | **0.22** |
| synergy | NP_size_max_nm | 0.46 | **0.13** | 0.49 | **0.13** | 0.02 | 0.01 |
| | NP_size_avg_nm | 0.58 | **0.20** | 0.53 | 0.15 | 0.18 | 0.11 |
| | shape | 0.47 | 0.13 | 0.68 | **0.21** | 0.33 | 0.19 |
| | method | 0.35 | 0.13 | 0.54 | 0.14 | 0.47 | **0.29** |
| | ZOI_drug_mm_or_MIC _µg_ml | 0.00 | 0.00 | 0.00 | 0.00 | 0.34 | **0.16** |
| | error_ZOI_drug_mm _or_MIC_µg_ml | 0.09 | 0.06 | 0.13 | 0.05 | 0.05 | 0.03 |
| | ZOI_NP_mm_ or_MIC_np_µg_ml | 0.57 | 0.16 | 0.57 | **0.17** | 0.16 | 0.10 |
| | error_ZOI_NP_mm_ or_MIC_np_µg_ml | 0.17 | 0.06 | 0.25 | 0.07 | 0.10 | 0.06 |
| | ZOI_drug_NP_mm_ or_MIC_drug_NP_µg_ml | 0.26 | 0.09 | 0.28 | 0.08 | 0.09 | 0.07 |
| | error_ZOI_drug_NP_ mm_or_MIC_drug_NP_µg_ml | 0.11 | 0.06 | 0.08 | 0.03 | 0.07 | 0.05 |
| | fold_increase_ in_antibacterial_activity | 0.04 | 0.00 | 0.02 | 0.00 | 0.03 | 0.00 |
| | zeta_potential_mV | 0.09 | 0.03 | 0.10 | 0.02 | 0.09 | 0.05 |
| | MDR | 0.47 | **0.21** | 0.49 | 0.19 | 0.00 | 0.00 |
| | FIC | 0.08 | 0.02 | 0.06 | 0.02 | 0.05 | 0.03 |
| | effect | 0.21 | 0.10 | 0.25 | 0.09 | 0.00 | 0.00 |
| | time_hr | 0.37 | **0.13** | 0.43 | 0.16 | 0.03 | 0.02 |
| | coating_with_ antimicrobial_peptide_polymers | 0.44 | **0.18** | 0.32 | 0.11 | 0.00 | 0.00 |
| | combined_MIC | 0.00 | 0.00 | 0.00 | 0.00 | 0.00 | 0.00 |
| | peptide_MIC | 0.00 | 0.00 | 0.00 | 0.00 | 0.00 | 0.00 |
| | viability_% | 0.00 | 0.00 | 0.00 | 0.00 | 0.01 | 0.01 |
| | viability_error | 0.00 | 0.00 | 0.00 | 0.00 | 0.01 | 0.01 |

Table 10: Extraction metrics for all columns of nanomag dataset. Baseline model and single agent methods are presented.

| Dataset | Columns | LLM FROM PDF | | LLM FROM JPEG | | SINGLE AGENT | |
|---|---|---|---|---|---|---|---|
| | | Precision | F1 | Precision | F1 | Precision | F1 |
| | name | 0.02 | 0.00 | 0.00 | 0.00 | 0.00 | 0.00 |
| | np_core | 0.32 | 0.21 | 0.41 | **0.26** | 0.22 | 0.16 |
| | np_shell | 0.29 | 0.21 | 0.32 | **0.24** | 0.07 | 0.06 |
| | core_shell_formula | 0.15 | 0.09 | 0.29 | **0.19** | 0.07 | 0.06 |
| | np_shell_2 | 0.71 | 0.46 | 0.78 | **0.53** | 0.00 | 0.00 |
| | np_hydro_size | 0.06 | 0.05 | 0.06 | 0.04 | 0.02 | 0.02 |
| | xrd_scherrer_size | 0.05 | 0.03 | 0.08 | 0.05 | 0.02 | 0.01 |
| | emic_size | 0.16 | 0.10 | 0.18 | **0.14** | 0.09 | 0.09 |
| | space_group_core | 0.30 | 0.20 | 0.30 | **0.24** | 0.02 | 0.02 |
| | space_group_shell | 0.38 | 0.26 | 0.35 | **0.29** | 0.02 | 0.02 |
| | squid_sat_mag | 0.19 | 0.12 | 0.20 | **0.16** | 0.14 | 0.12 |
| nanomag | squid_rem_mag | 0.27 | 0.17 | 0.24 | **0.19** | 0.02 | 0.02 |
| | exchange_bias_shift_Oe | 0.02 | 0.01 | 0.00 | 0.00 | 0.01 | 0.01 |
| | vertical_loop_shift_M_vsl_emu_g | 0.09 | 0.05 | 0.04 | 0.02 | 0.00 | 0.00 |
| | hc_kOe | 0.01 | 0.01 | 0.01 | 0.01 | 0.00 | 0.00 |
| | squid_h_max | 0.16 | 0.09 | 0.17 | 0.09 | 0.00 | 0.00 |
| | zfc_h_meas | 0.00 | 0.00 | 0.00 | 0.00 | 0.00 | 0.00 |
| | instrument | 0.15 | **0.11** | 0.05 | 0.05 | 0.00 | 0.00 |
| | fc_field_T | 0.10 | 0.08 | 0.06 | 0.04 | 0.07 | 0.04 |
| | squid_temperature | 0.32 | 0.19 | 0.34 | **0.22** | 0.09 | 0.08 |
| | coercivity | 0.20 | **0.12** | 0.15 | 0.11 | 0.02 | 0.02 |
| | htherm_sar | 0.00 | 0.00 | 0.01 | 0.01 | 0.01 | 0.01 |
| | mri_r1 | 0.09 | 0.07 | 0.17 | **0.15** | 0.04 | 0.02 |
| | mri_r2 | 0.13 | 0.09 | 0.18 | **0.14** | 0.06 | 0.05 |

## 9.5 Prompts

**Benzimidazole antibiotics**

**system_prompt** = "You are a domain-specific chemical information extraction assistant. You specialize in chemistry of small molecules. In particular, your area is antibiotics and their properties."

**user_prompt** = "Your task is to extract **every** mention of MIC or pMIC measurements against Staphylococcus aureus and Escherichia coli bacteria for **ALL** benzimidazole antibiotics from a scientific article and output a **JSON array** of objects **only** (no markdown, no commentary, no extra text).

Fields for each object:

- 'compound_id' (string): ID of a molecule within the article, as cited in the text, e.g. '"5a"', '"Compound 3"'.
- 'smiles' (string): full SMILES representation of a benzimidazole antibiotic.
- 'target_type' (string): type of measurement, either '"MIC"' or '"pMIC"', exactly as stated.
- target_relation' (string): one of '"="', '"<"', or '">"'. If no relation symbol is shown, use '"="'.
- 'target_value' (number): the numeric value of MIC/pMIC (without quotes).
- 'target_units' (string): MIC units, e.g. '"$\mu$g/mL"', '"mg/L"', etc.
- 'bacteria' (string): the organism against which MIC/pMIC was measured, named exactly as in the text.

Extraction rules:

1. Extract **each** MIC/pMIC mention as a separate object. If multiple MIC/pMIC are reported for the same compound against different bacteria, list them as separate entries.
2. Do **not** filter, group, summarize, or deduplicate. Include repeated mentions and duplicates if they occur in different contexts.
3. If a range is given (e.g., "2–8 $\mu$g/mL"), leave it as a range.
4. If a molecule is fully depicted in a figure, write it as a SMILES string. If a molecule is depicted as a scaffold and residues separately in different places of an article, connect them by compound ID into one molecule and write it as a single SMILES string.
5. Extract only measurements with Staphylococcus aureus and Escherichia coli. Record full names, abbreviations, or any related taxonomic identifiers of bacteria.
6. If you cannot find a required field for an object, re-check the context; if it's still absent, set that field's value to '"NOT_DETECTED"'.
7. The example of JSON below shows only two extracted samples, however your output should contain **all** MIC or pMIC measurements of benzimidazole antibiotics present in the article.

Output **must** be a single JSON array, like:
[ {
"compound_id": "11h",
"smiles": "O=C(OCC)C1=C(N(C(=O)N(C1C2=C(C=CS2)C)[H])
[H])C[N]3C=NC4=C3C=C(C=C4)[N+](=O)[O-]",
"target_type": "MIC",
"target_relation": "<",
"target_value": 1,
"target_units": "mmol/l",
"bacteria": "methicillin-susceptible S. aureus" },
{
"compound_id": "5a",
"smiles": "CCN1C=C(C(=O)C2=CC(=C(C=C21)N3CCN(CC3)
C4=NC=CC(=N4)N)F)C(=O)O",

"target_type": "pMIC",
"target_relation": "<",
"target_value": 2,
"target_units": "$\mu$g/mL",
"bacteria": "Escherichia coli" }]

**Oxazolidinone antibiotics**

**system_prompt** = "You are a domain-specific chemical information extraction assistant. You specialize in chemistry of small molecules. In particular, your area is antibiotics and their properties."

**user_prompt** = "Your task is to extract **every** mention of MIC or pMIC values for oxazolidinone antibiotics from a scientific article and output a **JSON array** of objects **only** (no markdown, no commentary, no extra text).

Fields for each object:

- 'compound_id' (string): ID of a molecule within the article, as cited in the text, e.g. '"5a"', '"Compound 3"'.
- 'smiles' (string): full SMILES representation of an oxazolidinone antibiotic.
- 'target_type' (string): type of measurement, either '"MIC"' or '"pMIC"', exactly as stated.
- 'target_relation' (string): one of '"="', '"<"', or '">"'. If no relation symbol is shown, use '"="'.
- 'target_value' (number): the numeric value of MIC/pMIC (without quotes).
- 'target_units' (string): e.g. '"$\mu$g/mL"', '"mg/L"', etc.
- 'bacteria' (string): the organism against which MIC/pMIC was measured, named exactly as in the text. Record full names, abbreviations, or any related taxonomic identifiers of bacteria.

Extraction rules:

1. Extract **each** MIC or pMIC mention as a separate object.
2. Do **not** filter, group, summarize, or deduplicate. Include repeated mentions and duplicates if they occur in different contexts.
3. If a range is given (e.g., "2–8 $\mu$g/mL"), leave it as a range.
4. If a molecule is fully depicted in a figure, write it as a SMILES string. If a molecule is depicted as a scaffold and residues separately in different places of an article, connect them by compound ID into one molecule and write it as a single SMILES string.
5. If multiple measurement types appear for the same compound and bacterium (e.g., $MIC_{50}$, $MIC_{90}$), extract each separately.
6. If you cannot find a required field for an object, re-check the context; if it's still absent, set that field's value to '"NOT_DETECTED"'.
7. The example of JSON below shows only two extracted samples, however your output should contain **all** MIC or pMIC measurements of oxazolidinone antibiotics present in the article.

Output **must** be a single JSON array, like:
[ {
"compound_id": "12b",
"smiles": "CC1=CC=C(C=C1)C(=O)Nc2ccc(cc2)C
(=O)N3CCCCC3=O",
"target_type": "MIC",
"target_relation": "<",
"target_value": 1,
"target_units": "mmol/l",
"bacteria": "methicillin-susceptible S. aureus" },
{
"compound_id": "5a",
"smiles": "CC1=CC=CC=C1N2C=NC3=CC=CC=C23",

```
"target_type": "MIC",
"target_relation": "=",
"target_value": 2,
"target_units": "µg/mL",
"bacteria": "Escherichia coli" } ]"
```

**Cocrystals**

**system_prompt** = "You are a domain-specific chemical information extraction assistant. You specialize in the chemistry of cocrystals and their properties. Your area of expertise includes analyzing cocrystals, their components, and photostability changes."

**user_prompt** = "Your task is to extract **every** mention of photostability for co-crystals from a scientific article, and output a **JSON array** of objects **only**(no markdown, no commentary, no extra text).

Fields for each object:

- 'name_cocrystal' (string): name of cocrystal, as cited in the text, e.g. '"CAR-HCT"', '"DMZ-SAC"'
- 'ratio_cocrystal' (string): molar ratio of the cocrystal components, e.g., '"2:1"', '"0.5:1".
- 'name_drug' (string): name of the drug in the cocrystal as cited in the text, e.g. '"Carvedilol"', '"Epalrestat"'.
- 'SMILES_drug' (string): full SMILES representation of drug.
- 'name_coformer' (string): name of the coformer in the cocrystal as cited in the text, e.g. '"Saccharin"', '"Oxalic acid"'.
- 'SMILES_coformer' (string): full SMILES representation of coformer.
- 'photostability_change' (string): one of '"decrease"', '"does not change"', or '"increase"'. Trend of photostability for both the cocrystal and the drug, indicating how their stability changes over time.

Extraction rules:

1. Extract **each** photostability mention as a separate object.
2. Do **not** filter, group, summarize, or deduplicate. Include repeated mentions and duplicates if they occur in different contexts.
3. If multiple polymorphic forms (e.g., CBZ-SAC Form I, CBZ-SAC Form II) appear for the same drug and coformer in the same ratio, extract each separately.
4. If you cannot find a required field for an object, re-check the context; if it's still absent, set that field's value to '"NOT_DETECTED"'.
5. The example of JSON below shows only two extracted samples, however your output should contain **all** mentions of photostability for co-crystals present in the article.

Output **must** be a single JSON array, like:
```
[ {
"name_cocrystal": "CAR-HCT",
"ratio_cocrystal": "2:1",
"name_drug": "Carvedilol",
"SMILES_drug": "C1=CC(=C(C=C1O)O)C=CC2=CC(=CC(=C2)O)O",
"name_coformer": "Saccharin",
"SMILES_coformer": "O=C(O)CC(O)C(=O)O",
"photostability_change": "decrease" }, {
"name_cocrystal": "DMZ-SAC",
"ratio_cocrystal": "0.5:1",
"name_drug": "Epalrestat",
"SMILES_drug": "C1=CC(=C(C=C1O)O)C=CC2=CC(=CC(=C2)O)O",
"name_coformer": "Oxalic acid",
"SMILES_coformer": "C(=C/C(=O)O)
```

C(=O)O",
"photostability_change": "does not change" } ]"

**Complexes**

**system_prompt** = "You are a domain-specific chemical information extraction assistant. You specialize in the chemistry of organometallic complexes and their properties."

**user_prompt** = "Your task is to extract **every** mention of organometallic complexes and chelate ligands from scientific article, and output a **JSON array** of objects **only** (no markdown, no commentary, no extra text).

Fields for each object:

- 'compound_id' (string): ID of a complex within the article, as cited in the text, e.g. '"L3"', '"A31"'.

- 'compound_name' (string): abbreviated or full name of the complex or ligand as cited in the text, e.g. '"DOTA"', '"tebroxime"'.

- 'SMILES' (string): full SMILES representation of ligand environment or single ligand. If a complete organometallic complex is shown, extract all ligand structures without mentioning the metal (e.g., "COc1cc(C=CC([O-])CC([O-])CC([O-])C=Cc2ccc(O)c(OC)c2)ccc1O. [C-]#[O+].[C-]#[O+].[C-]#[O+].[OH-]"). For a chelate ligand without a complete organometallic complex, extract only that ligand's structure (e.g., 'O=C(O)CN(CCN(CC(CC(=O)O)CC(=O)O)CCN(CC(=O)O)CC(=O)O').

- 'SMILES_type' (string): one of '"ligand"' or '"environment"'. "environment" refers to the entire organometallic complex, including one or more ligands and a metal atom.

- 'target_value' (number): the numeric value of logarithms of thermodynamic stability constants lgK or logK (without quotes).

Extraction rules:

1. Extract **each** mention of 'target_value' (lgK or logK) as a separate object.

2. Do **not** filter, group, summarize, or deduplicate. Include repeated mentions and duplicates if they occur in different contexts.

3. If a molecule is fully depicted in a figure, write it as a SMILES string. If a molecule is depicted as a scaffold and residues separately in different places of an article, connect them by compound ID or name into one molecule and write it a single SMILES string.

4. If multiple thermodynamic stability constants appear for the same complex or ligand extract each separately.

5. Extract only structures that comply with these rules:
   - The complexes must contain **Ga** as the metal or the ligands must belong to complexes of that metal.
   - The complete molecular structure shall be given without errors in it or identifiers.
   - Compounds must contain more than one carbon (exclude CO, Me).
   - Compounds must not contain polymeric structures, attached biomolecules or carboranes, undefined radicals, undeciphered designations (e.g., amino acids) beyond the simplest abbreviations (i.e., Me, Et, Pr, Bu, Ph, Ac), names of radicals instead of their structure, or incomplete indication of the ligand structure (e.g., L = P, N).
   - Compounds must not be reaction intermediate or precursor.

6. If you cannot find a required field for an object, re-check the context; if it's still absent, set that field's value to '"NOT_DETECTED"'.

7. The example of JSON below shows only two extracted samples, however your output should contain **all** mentions of organometallic complexes and / or chelate ligands present in the article.

Output **must** be a single JSON array, like:
[ {

"compound_id": "L3",
"compound_name": "DOTA",
"SMILES": "O=C(O)CN(CCN(CC(=O)O)CC(=O)O)CC(=O)O",
"SMILES_type": "ligand",
"target": 21.3 }, {
"compound_id": "A31",
"compound_name": "tebroxime",
"SMILES": "[C-]#[N+]CC(C)(C)OC.[C-]#[N+]CC(C)(C)OC.[C-]#[N+]CC(C)(C)OC.[C-]#[N+]CC(C)(C)OC.[C-]#[N+]CC(C)(C)OC.[C-]#[N+]CC(C)(C)OC",
"SMILES_type": "environment",
"target": 17.9 } ]"

**Nanozymes**

**system_prompt** = "You are a domain-specific chemical information extraction assistant. You specialize in nanozymes."

**user_prompt** = "Your task is to extract **every** mention of experiments for **ALL** nanozymes from a scientific article and output a **JSON array** of objects **only** (no markdown, no commentary, no extra text).

Fields for each object:

- 'formula' (string): the chemical formula of the nanozyme, e.g. "Fe3O4", "CuO", etc.
- 'activity' (string): catalytic activity type, typically "peroxidase", "oxidase", "catalase", "laccase", or other.
- 'syngony' (string): the crystal unit of the nanozyme, e.g. "cubic", "hexagonal", "tetragonal", "monoclinic", "orthorhombic", "trigonal", "amorphous", "triclinic".
- 'length' (number): the length of the nanozyme particle in nanometers.
- 'width' (number): the width of the nanozyme particle in nanometers.
- 'depth' (number): the depth of the nanozyme particle in nanometers.
- 'surface' (string): the molecule on the surface of the nanozyme, e.g., "naked", "poly(ethylene oxide)", "poly(N-Vinylpyrrolidone)", "Tetrakis(4-carboxyphenyl)porphine", or other.
- 'km_value' (number): the Michaelis constant value for the nanozyme.
- 'km_unit' (string): the unit for the Michaelis constant, e.g., "mM", etc.
- 'vmax_value' (number): the molar maximum reaction rate value.
- 'vmax_unit' (string): the unit for the maximum reaction rate, e.g., "$\mu$mol/min", "mol/min", etc.
- 'reaction_type' (string): the reaction type involving the substrate and co-substrate, e.g., "TMB + H2O2", "H2O2 + TMB", "TMB", "ABTS + H2O2", "H2O2", "OPD + H2O2", "H2O2 + GSH", or other.
- 'c_min' (number): the minimum substrate concentration in catalytic assays in mM.
- 'c_max' (number): the maximum substrate concentration in catalytic assays in mM.
- 'c_const' (number): the constant co-substrate concentration used during assays.
- 'c_const_unit' (string): the unit of measurement for co-substrate concentration.
- 'ccat_value' (number): the concentration of the catalyst used in assays.
- 'ccat_unit' (string): the unit of measurement for catalyst concentration.
- 'ph' (number): the pH level at which experiments were conducted.
- 'temperature' (number): the temperature in Celsius during the study.

Extraction rules:

1. Extract **each** nanozyme mention as a separate object.
2. Do **not** filter, group, summarize, or deduplicate. Include repeated mentions and duplicates if they occur in different contexts.

3. If you cannot find a required field for an object, re-check the context; if it's still absent, set that field's value to '"NOT_DETECTED"'.

4. The example of JSON below shows only two extracted samples, however your output should contain **all** nanozymes present in the article.

Output **must** be a single JSON array, like:
[ {
"formula": "Fe3O4",
"activity": "peroxidase",
"syngony": "cubic",
"length": 10,
"width": 10,
"depth": 2.5,
"surface": "naked",
"km_value": 0.2,
"km_unit": "mM",
"vmax_value": 2.5,
"vmax_unit": "μmol/min",
"reaction_type": "TMB + H2O2",
"c_min": 0.01,
"c_max": 1.0,
"c_const": 1.0,
"c_const_unit": "mM",
"ccat_value": 0.05,
"ccat_unit": "mg/mL",
"ph": 4.0,
"temperature": 25 }, {
"formula": "CeO2",
"activity": "oxidase",
"syngony": "cubic",
"length": 5,
"width": 5,
"depth": 200,
"surface": "poly(ethylene oxide)",
"km_value": 54.05,
"km_unit": "mM",
"vmax_value": 7.88,
"vmax_unit": "10-8 M s-1",
"reaction_type": "TMB",
"c_min": 0.02,
"c_max": 0.8,
"c_const": 800,
"c_const_unit": "$\mu$M",
"ccat_value": 0.02,
"ccat_unit": "mg/mL",
"ph": 5.5,
"temperature": 37 } ]"

**Nanomag**

**system_promt** = "You are a domain-specific chemical information extraction assistant. You specialize in nanomaterials characterization, specifically in magnetic nanoparticles and their physical properties."

**user_prompt** = "Your task is to extract **every** mention of magnetic properties for **ALL** nanoparticles from a scientific article and output a **JSON array** of objects **only** (no markdown, no commentary, no extra text).

Fields for each object:

- 'name' (string): material name (e.g., BFO, cobalt irin oxide and bismuth ferrite etc.).

- 'np_core' (string): composition of material core (e.g., Gd2O3, Fe1Fe2O4 etc.).
- 'np_shell' (string): composition of material shell (e.g., chitosan, Au1 etc.).
- 'core_shell_formula' (string): sometimes nanoparticle composition is represented as one formula containing both core and shell parts; core and shell materials are typically separated by a delimiter such as -, /, @, or |, e.g. Cr2O3-Co.
- 'np_shell_2' (string): first additional shell layer if present (e.g., PEG-5000, Curcumin etc.).
- 'np_hydro_size' (number): size of nanoparticles in solution obtained by dynamic light scattering (DLS) or similar, in nanometers (nm).
- 'xrd_scherrer_size' (number): crystal size calculated from x-ray diffraction, usually represented in figures, in nanometers (nm).
- 'emic_size' (number): size measured by electron microscopy, usually represented in figures, in nanometers (nm).
- 'space_group_core' (string): space groups of core material (e.g., fd-3m, p4/mmm, etc.).
- 'space_group_shell' (string): space groups of shell material (e.g., fd-3m, p4/mmm, etc.).
- 'squid_sat_mag' (number): saturation magnetization (Ms, Bs) in emu/g.
- 'exchange_bias_shift_Oe' (number): exchange bias (Heb, exchange bias effect) in Oersted (Oe).
- 'vertical_loop_shift_M_vsl_emu_g' (number): vertical loop shift (vertical bias) in emu/g.
- 'hc_kOe' (number): coercivity (Hc, coercive force) in Oersted (Oe).
- 'squid_h_max' (number): maximum magnetic field in kOe.
- 'zfc_h_meas' (number): measurement field for ZFC in kOe.
- 'instrument' (string): experimental instrument (e.g., Quantum Design 7 T SQUID magnetometer, Seifert XRD 3000P, etc.).
- 'fc_field_T' (number): FC field in Tesla (T).
- 'squid_temperature' (number): squid temperature in Kelvin.
- 'coercivity' (number): coercivity (Hc) in kOe.
- 'htherm_sar' (number): specific absorption rate (SAR) in W/g.
- 'mri_r1' (number): MRI relaxation rate r1 in mM-1·s-1.
- 'mri_r2' (number): MRI relaxation rates r2 in mM-1·s-1.

Extraction rules:

1. Extract **each** nanoparticle mention as a separate object.
2. Do **not** filter, group, summarize, or deduplicate. Include repeated mentions and duplicates if they occur in different contexts.
3. If you cannot find a required field for an object, re-check the context; if it's still absent, set that field's value to '"NOT_DETECTED"'.
4. If the original unit of coercivity or exchange bias is different, it must be converted into Oe: 1T = 1000 Oe, 1 mT = 10000 Oe, 1kOe = 1000 Oe.
5. Do not remove or alter the negative (-) or positive (+) signs for exchange bias and vertical loop shift. If the article does not explicitly state the sign, assume it is (+) by default.
6. The example of JSON below shows only one extracted sample, however your output should contain entries for **all** magnetic nanoparticles present in the article.

Output **must** be a single JSON array, like:
[ {
"name": "Bismuth Ferrite",
"np_core": "BiFeO3",
"np_shell": "chitosan",
"core_shell_formula": "BiFeO3-chitosan",

"np_shell_2": "PEG-5000",
"np_hydro_size": 120,
"xrd_scherrer_size": 45,
"emic_size": 50,
"space_group_core": "R3c",
"space_group_shell": "P2_1",
"squid_sat_mag": 40.5,
"squid_rem_mag": 22.1,
"exchange_bias_shift_Oe": 180,
"vertical_loop_shift_M_vsl_emu_g": 5.6,
"hc_kOe": 3.2,
"squid_h_max": 5.0,
"zfc_h_meas": 1.5,
"instrument": "Quantum Design 7 T SQUID magnetometer",
"fc_field_T": 0.1,
"squid_temperature": 300,
"coercivity": 3.5,
"htherm_sar": 1.2,
"mri_r1": 4.5,
"mri_r2": 5.3, } ]"

**Synergy**

**system_prompt** = "You are a domain-specific chemical information extraction assistant. You specialize in antimicrobial drug nanoparticle synergy."

**user_prompt** = "Your task is to extract **every** mention of nanoparticle properties, drug details, and their synergistic antibacterial effects from a scientific article, and output a **JSON array** of objects **only** (no markdown, no commentary, no extra text).

Fields for each object:

- 'NP' (string): nanoparticle name as cited in the text, e.g. , "Ag", "Au".

- 'bacteria' (string): bacterial strain tested, e.g., "Escherichia coli".

- 'strain' (string): specific strain identifier for the bacteria tested as cited in the text, e.g., "ATCC 25922", "MTCC 443".

- 'NP_synthesis' (string): method by which the nanoparticles were synthesized, e.g., "chemical synthesis", "hydrothermal synthesis".

- 'drug' (string): name of the conventional antibiotic or other antimicrobial drug used in combination with the nanoparticles, e.g., "Ampicillin", "Ciprofloxacin".

- 'drug_dose_$\mu$g_disk' (number): specific dosage or concentration of the drug applied, primarily used for methods like disc diffusion assays, typically measured in micrograms per disk.

- 'NP_concentration_$\mu$g_ml' (number): concentration of the nanoparticle used in the antibacterial assay, e.g., for MIC, ZOI, or viability studies, typically measured in micrograms per milliliter.

- 'NP_size_min_nm' (number): the smallest recorded size of the nanoparticle particles as determined by characterization techniques, measured in nanometers.

- 'NP_size_max_nm' (number): the largest recorded size of the nanoparticle particles as determined by characterization techniques, measured in nanometers.

- 'NP_size_avg_nm' (number): the average size of the nanoparticle particles, typically based on measurements from techniques like TEM or DLS, measured in nanometers.

- 'shape' (string): observed morphology or physical shape of the nanoparticle particles, e.g., "spherical", "rod-shaped", "cubic", "irregular", "nanosheets".

- 'method' (string): specific experimental technique employed to assess the antibacterial efficacy or interaction, e.g., "MIC", "disc_diffusion", "well_diffusion", "broth microdilution", "time-kill assay".

- 'ZOI_drug_mm_or_MIC_μg_m' (number): quantitative measure of antibacterial activity for the drug alone. This will be the diameter of the ZOI in millimeters for disc diffusion assays, or the MIC value in micrograms per milliliter for methods like broth microdilution.

- 'error_ZOI_drug_mm_or_MIC_μg_ml' (number): uncertainty or variability associated with the antibacterial activity measurement for the drug alone, often represented as the standard deviation.

- 'ZOI_NP_mm_or_MIC_np_μg_ml' (number): The quantitative measure of antibacterial activity for the nanoparticle alone. This will be the ZOI diameter in millimeters or the MIC value in micrograms per milliliter.

- 'error_ZOI_NP_mm_or_MIC_np_μg_ml' (number): uncertainty or variability associated with the antibacterial activity measurement for the nanoparticle alone.

- 'ZOI_drug_NP_mm_or_MIC_drug_NP_μg_ml' (number): quantitative measure of antibacterial activity for the combination of the drug and the nanoparticle. This will be the ZOI diameter in millimeters or the MIC value in micrograms per milliliter.

- 'error_ZOI_drug_NP_mm_or_MIC_drug_NP_μg_ml' (number): uncertainty or variability associated with the antibacterial activity measurement for the drug + nanoparticle combination.

- 'fold_increase_in_antibacterial_activity' (number): numerical value indicating how much more effective the combination of the drug and nanoparticle is compared to the most effective component used individually.

- 'zeta_potential_mV' (number): electrokinetic potential of the nanoparticle surface, measured in millivolts. It is an indicator of the surface charge and stability of the nanoparticles in suspension.

- 'MDR' (string): indicator of whether the bacterial strain tested exhibits multidrug resistance, e.g., "Yes", "No", "Resistant", "Susceptible".

- 'FIC' (number): Fractional Inhibitory Concentration index value, calculated to assess the interaction between the drug and nanoparticle. Values help determine if the interaction is synergistic (<0.5), additive (0.5-1.0), indifferent (1.0-4.0), or antagonistic (>4.0).

- 'effect' (string): qualitative description of the interaction between the drug and nanoparticle based on the FIC index, e.g., "synergistic", "additive", "antagonistic", "indifferent".

- 'time_hr' (number): duration of exposure of the bacterial cells to the antibacterial agents during the experiment, specified in hours.

- 'coating_with_antimicrobial_peptide_polymers' (string): indicates whether the nanoparticles were modified with a coating of antimicrobial peptides or polymers to enhance their activity or targeting, e.g., "yes", "no", specifies the coating material.

- 'combined_MIC' (number): Minimum Inhibitory Concentration observed for the combination of an antimicrobial peptide / polymer coating and the nanoparticle, in micrograms per milliliter if applicable.

- 'peptide_MIC' (number): Minimum Inhibitory Concentration of the antimicrobial peptide Used in isolation, in micrograms per milliliter if applicable.

- 'viability_%' (number): percentage of bacterial cells that survive or remain viable after being exposed to the nanoparticle, drug, or combination for a specific time period.

- 'viability_error' (number): associated error or standard deviation for the bacterial viability percentage measurement.

Extraction rules:

1. Extract **each** nanoparticles mention as a separate object.

2. Do **not** filter, group, summarize, or deduplicate. Include repeated mentions and duplicates if they occur in different contexts.

3. If you cannot find a required field for an object, re-check the context; if it's still absent, set that field's value to '"NOT_DETECTED"'.

4. The example of JSON below shows only two extracted samples, however your output should contain **all** nanoparticles present in the article.

Output **must** be a single JSON array, like:
[ {
"NP": "Ag",
"bacteria": "Pseudomonas aeruginosa",
"strain": "ATCC 27853",
"NP_synthesis": "Green synthesis using Gloeophyllum striatum",
"drug": "Ampicillin",
"drug_dose_µg_disk": 16.0,
"NP_concentration_µg_ml": 32.0,
"NP_size_min_nm": 10.0,
"NP_size_ma_nm": 40.0,
"NP_size_avg_nm": 20.0,
"shape": "spherical", "method": "MIC",
"ZOI_drug_mm_or_MIC_µg_ml": 16.0,
"error_ZOI_drug_mm_or_MIC_µg_ml": 1.40,
"ZOI_NP_mm_or_MIC_np_µg_ml": 32.0,
"error_ZOI_NP_mm_or_MIC_np_µg_ml": 2.43,
"ZOI_drug_NP_mm_or_MIC_drug_NP_µg_ml": 8.0,
"error_ZOI_drug_NP_mm_or_MIC_drug_NP_µg_ml": 1.50,
"fold_increase_in_antibacterial_activity": 2.0,
"zeta_potential_mV": -34.0,
"MDR": "R",
"FIC": 0.5,
"effect": "synergistic",
"time_hr": 24.0,
"coating_with_antimicrobial_peptide_polymers": "AP Lysozyme hen egg-white",
"combined_MIC": 12,
"peptide_MIC": 400,
"viability_%": 87.0,
"viability_error": 2.40 }, {
"NP": "Au",
"bacteria": "Escherichia coli",
"strain": "BJ915",
"NP_synthesis": "purchased from Jinke Chemical Co",
"drug": "Colistin",
"drug_dose_µg_disk": 10.0,
"NP_concentration_µg_ml": 25.0,
"NP_size_min_nm": 2.1,
"NP_size_max_nm": 2.9,
"NP_size_avg_nm": 2.5,
"shape": "cubic",
"method": "MBC",
"ZOI_drug_mm_or_MIC_µg_ml": 4.0,
"error_ZOI_drug_mm_or_MIC_µg_ml": 0.30,
"ZOI_NP_mm_or_MIC_np_µg_ml": 12.50,
"error_ZOI_NP_mm_or_MIC_np_µg_ml": 0.87,
"ZOI_drug_NP_mm_or_MIC_drug_NP_µg_ml": 6.25,
"error_ZOI_drug_NP_mm_or_MIC_drug_NP_µg_ml": 0.27,
"fold_increase_in_antibacterial_activity": 1.16,
"zeta_potential_mV": 14.0,
"MDR": "R",
"FIC": 0.75,
"effect": "P",
"time_hr": 24.0,
"coating_with_antimicrobial_peptide_polymers":       "4,6-diamino-2-pyrimidinethiol    +    1,1-dimethylbiguanide",

"combined_MIC": 4.0,
"peptide_MIC": 13.20,
"viability_%": 23.0,
"viability_error": 2.25 } ]"

**Seltox**

**system_prompt** = @You are a domain-specific chemical information extraction assistant. You specialize in antimicrobial nanoparticles."

**user_prompt** = "Your task is to extract information for **ALL** antimicrobial nanoparticles from a scientific article and output a **JSON array** of objects **only** (no markdown, no commentary, no extra text).

Fields for each object:

- 'np' (string): Nanoparticle name (e.g., "Ag", "Au", "ZnO").
- 'coating' (string): Surface coating/modification ("1" for coating, "0" for none).
- 'bacteria' (string): Bacterial strain tested (e.g., "Escherichia coli", "Staphylococcus aureus").
- 'mdr' (number): Multidrug-resistant strain indicator, one of 1 or 0 (1 for multidrug-resistant, 0 for not multidrug-resistant).
- 'strain' (string): Specific strain identifier (e.g., "ATCC 25922").
- 'np_synthesis' (string): Synthesis method (e.g., "green_synthesis", "chemical_synthesis", or specific details like "Green synthesis using Pimpinella anisum").
- 'method' (string): Assay type (e.g., "MIC", "ZOI", "MBC", "MBEC").
- 'mic_np_μg_ml' (number): Minimum Inhibitory Concentration (MIC) in $\mu$g/mL.
- 'concentration' (number): Concentration for Zone of Inhibition (ZOI) in $\mu$g/mL.
- 'zoi_np_mm' (number): Zone of Inhibition in mm.
- 'np_size_min_nm' (number): Minimum nanoparticle size in nm.
- 'np_size_max_nm' (number): Maximum nanoparticle size in nm.
- 'np_size_avg_nm' (number): Average nanoparticle size in nm.
- 'shape' (string): Morphology (e.g., "spherical", "triangular").
- 'time_set_hours' (number): Experiment duration in hours.
- 'zeta_potential_mV' (number): Surface charge in mV.
- 'solvent_for_extract' (string): Solvent used in green synthesis (e.g., "water", "ethanol").
- 'temperature_for_extract_C' (number): Temperature during extract preparation in °C.
- 'duration_preparing_extract_min' (number): Time to prepare extract in minutes.
- 'precursor_of_np' (string): Chemical precursor (e.g., "AgNO3").
- 'concentration_of_precursor_mM' (number): Precursor concentration in mM.
- 'hydrodynamic_diameter_nm' (number): Hydrodynamic size in nm.
- 'ph_during_synthesis' (number): pH of synthesis solution.

Extraction rules:

1. Extract solvents and precursors as strings without parsing into molecular components.
2. Extract **each** nanoparticle mention as a separate object.
3. Do **not** filter, group, summarize, or deduplicate. Include repeated mentions and duplicates if they occur in different contexts.
4. If you cannot find a required field for an object, re-check the context; if it's still absent, set that field's value to '"NOT_DETECTED"'.
5. The example of JSON below shows only two extracted samples, however your output should contain **all** nanoparticles present in the article.

Output **must** be a single JSON array, like:

[ {
"np": "Ag",
"coating": "0",
"bacteria": "Enterococcus faecalis",
"mdr": 0,
"strain": "ATCC 29212",
"np_synthesis": "Green synthesis using Ixora brachypoda",
"method": "MIC",
"mic_np_µg_ml": 32.0,
"concentration": 10,
"zoi_np_mm": 15,
"np_size_min_nm": 10.0,
"np_size_max_nm": 40.0,
"np_size_avg_nm": 20.0,
"shape": "spherical",
"time_set_hours": 24,
"zeta_potential_mV": -27.9,
"solvent_for_extract": "water",
"temperature_for_extract_C": 21.0,
"duration_preparing_extract_min": 1440,
"precursor_of_np": "AgNO3",
"concentration_of_precursor_mM": 1.0,
"hydrodynamic_diameter_nm": 55,
"ph_during_synthesis": 8.5 }, {
"np": "ZnO",
"coating": "0",
"bacteria": "Klebsiella pneumoniae",
"mdr": 1,
"strain": "K-36",
"np_synthesis": "Green synthesis using Phyllanthus emblica",
"method": "MIC",
"mic_np_µg_ml": 6.25,
"concentration": 64,
"zoi_np_mm": 12,
"np_size_min_nm": 20.0,
"np_size_max_nm": 20.0,
"np_size_avg_nm": 20.0,
"shape": "spherical",
"time_set_hours": 24.0,
"zeta_potential_mV": -32,
"solvent_for_extract": "methanol",
"temperature_for_extract_C": 60,
"duration_preparing_extract_min": 60,
"precursor_of_np": "Zn(NO3).6.H2O",
"concentration_of_precursor_mM": 10,
"hydrodynamic_diameter_nm": 30,
"ph_during_synthesis": 7.0 } ]"

**Cytotoxicity**

**system_prompt** = "You are a domain-specific chemical information extraction assistant. You specialize in cytotoxic nanoparticles."

**user_prompt** = "Your task is to extract information for **ALL** cytotoxic nanoparticles from a scientific article and output a **JSON array** of objects **only** (no markdown, no commentary, no extra text).

Fields for each object:

- 'material' (string): Composition of the nanoparticle/material tested (e.g., "SiO2", "Ag").

- 'shape' (string): Physical shape of the particle (e.g., "Sphere", "Rod").

- 'coat_functional_group' (string): Surface coating or functionalization (e.g., "CTAB", "PEG").
- 'synthesis_method' (string): Synthesis method (e.g., "Precipitation", "Commercial").
- 'surface_charge' (string): one of '"Negative"', '"Neutral"', or '"Positive"'. Reported surface charge.
- 'core_nm' (number): Primary particle size in nm.
- 'size_in_medium_nm' (number): Hydrodynamic size in biological medium in nm.
- 'hydrodynamic_nm' (number): Size in solution including coatings in nm.
- 'potential_mv' (number): Surface charge in solution in mV.
- 'zeta_in_medium_mv' (number): Zeta potential in medium in mV.
- 'no_of_cells_cells_well' (number): Cell density per well in the assay.
- 'human_animal' (string): one of "A" for Animal or "H" for Human. Origin of cells.
- 'cell_source' (string): Species/organism (e.g., "Rat", "Human").
- 'cell_tissue' (string): Tissue origin of the cell line (e.g., "Adrenal Gland", "Lung").
- 'cell_morphology' (string): Cell shape (e.g., "Irregular", "Epithelial").
- 'cell_age' (string): Developmental stage of cells (e.g., "Adult", "Embryonic").
- 'time_hr' (number): Exposure duration in hours.
- 'concentration' (number): Tested concentration of the material (unit-specific, e.g., $\mu$g/mL).
- 'test' (string): Cytotoxicity assay type (e.g., "MTT", "LDH").
- 'test_indicator' (string): Reagent measured (e.g., "TetrazoliumSalt" for MTT).
- 'viability_%' (number): Cell viability percentage relative to control.

Extraction rules:

1. If multiple values are reported (e.g., sizes), prioritize TEM-measured sizes for core_nm. For concentration, note unit context from article if ambiguous.

2. Error Handling: Prioritize table data over text; note assumptions for ambiguous data.

3. Viability Notes: For viability_percent, values >100% may indicate proliferation stimulation; extract as reported.

4. Extract **each** nanoparticle mention as a separate object.

5. Do **not** filter, group, summarize, or deduplicate. Include repeated mentions and duplicates if they occur in different contexts.

6. If you cannot find a required field for an object, re-check the context; if it's still absent, set that field's value to '"NOT_DETECTED"'.

7. The example of JSON below shows only two extracted samples, however your output should contain **all** nanoparticles present in the article.

Output **must** be a single JSON array, like:
[ {
"material": "SiO2",
"shape": "Rod",
"coat_functional_group": "PEG",
"synthesis_method": "Precipitation",
"surface_charge": "Negative",
"core_nm": 20.0,
"size_in_medium_nm": 25.0,
"hydrodynamic_nm": 30.0,
"potential_mv": -15.0,
"zeta_in_medium_mv": -10.0,
"no_of_cells_cells_well": 5000.0,
"human_animal": "H",

"cell_source": "Human",
"cell_tissue": "Lung",
"cell_morphology": "Epithelial",
"cell_age": "Adult",
"time_hr": 24.0,
"concentration": 100.0,
"test": "MTT",
"test_indicator": "TetrazoliumSalt",
"viability_%": 85.0 }, {
"material": "Fe3O4",
"shape": "Sphere",
"coat_functional_group": "Dextran",
"synthesis_method": "Thermal Decomposition",
"surface_charge": "Positive",
"core_nm": 10.0,
"size_in_medium_nm": 15.0,
"hydrodynamic_nm": 18.0,
"potential_mv": -30.0,
"zeta_in_medium_mv": -15.0,
"no_of_cells_cells_well": 10000.0,
"human_animal": "A",
"cell_source": "Dog",
"cell_tissue": "Kidney",
"cell_morphology": "Epithelial",
"cell_age": "Adult",
"time_hr": 24.0,
"concentration": 300.0,
"test": "MTT",
"test_indicator": "TetrazoliumSalt",
"viability_%": 115.09 } ]"

