# OpenReview forum: "ChemX: A Collection of Chemistry Datasets for Benchmarking Automated Information Extraction"
_NeurIPS.cc/2025/Datasets_and_Benchmarks_Track — NeurIPS 2025 Datasets and Benchmarks Track poster_

### Official Review · Reviewer_eFPF · 2025-06-07

**Rating:** 5
**Confidence:** 4

**Summary:**

Automated information extraction in chemistry is an important domain-application of ML, which can help speed up data-driven scientific discoveries. This work presents ChemX, which contains 10 benchmarking datasets encompassing various properties of small molecules and nanomaterials. ChemX was manually extracted from peer-reviewed publications and validated by domain experts.  Experiments were conducted to evaluate the performance of LLMs in extracting structured data from scientific articles.

**Additional Feedback:**

(a) Section 3.3: It reads like only 20% of the data was validated. If so, the quality of the remaining 80% data may be questionable.
(b) How to combine the results of multi-agents to obtain the final results?

**Dataset Code Accessibility:**

Yes

**Ethical Considerations:**

No, there are no or only very minor ethics concerns

**Final Justification:**

The authors' responses address my major concerns. Hence, I raise the score to 5.

**Limitations Weaknesses:**

It reads that the data does not contain the original text/table where the information was extracted. It is highly recommended to include them. It will be extremely valuable.

**Strengths Contributions:**

ChemX is more comprehensive than existing ones in including the properties of nanomaterials and small molecules. It is well documented and was cross-verification by domain experts. Provide prompt-based and agent-based evaluation of LLMs as the baseline for future studies.

---

> ### Author Rebuttal · Authors · 2025-07-30
>
> We sincerely thank the reviewer for their thoughtful and constructive feedback. Below, we address the main concerns and clarify key points raised in the review.
>
> 1. We appreciate the reviewer’s suggestion regarding the inclusion of original text and tables from which ChemX data was extracted. We have already included this meta-information into the small molecule datasets. The column indicating the source of information (e.g., table, figure) is labeled with the term "origin." For nanomaterials, this metadata is currently in progress. Although these datasets already include a substantial number of articles, we are actively working on further updates and plan to expand the collection in the future.
>
> Additional feedback:
>
> a. Thank you for pointing out the need for clarity on the validation procedure. As described in Section 3.3 and illustrated in Figure 4 of the manuscript, we employed a stratified cross-verification strategy to validate data quality. Specifically, approximately 20% of entries from each article were randomly sampled and manually verified. However, this represents a lower bound. In practice, whenever isolated errors were identified, all entries from the corresponding article were re-checked and corrected as needed. This cascading verification mechanism ensured that common and article-specific patterns of errors were systematically addressed.
> As shown in Figure 5, the overall error rate across datasets did not exceed 4%, indicating that the initial data quality was already high. We will revise the manuscript to better highlight that the validation process extended beyond the initial 20% sample where necessary, ensuring high integrity of the full dataset.
>
> b. We thank the reviewer for this question. In our experiments, outputs from specialized agents (e.g., NER, image, and table parsers) are coordinated by a controller that validates, aligns, and merges them into schema-compliant outputs. As demonstrated by nanoMINER, multi-agent setups can achieve high performance, but only when trained for a narrow domain. However, as shown in our results, such systems do not generalize easily across datasets. We see ChemX as a foundation for building and evaluating future multi-agent pipelines that can generalize across diverse chemical domains, enabling more robust and reusable scientific extraction tools. We will clarify this vision in the revised version.
> Regarding broader integration of agent-based approaches for automated extraction (e.g., ensemble-like methods), we acknowledge the potential for future work. For instance, combining outputs from heterogeneous agents could theoretically improve robustness. However, as our experiments show, current single-agent performance is often too weak or too narrow to enable effective ensemble strategies without further advances in generalization.
>
> We sincerely thank the reviewer for highlighting the strengths of our work and kindly ask to consider elevating the score.

---

### Official Review · Reviewer_ozaP · 2025-06-29

**Rating:** 5
**Confidence:** 4

**Summary:**

ChemX is a collection of 10 benchmark datasets that have been manually annotated and validated by experts, covering two main domains:

- Small molecules
- Nanomaterials

These datasets are derived from over 1,500 peer-reviewed scientific articles. The data is presented in various formats, including text, tables, and images, and covers a wide range of physicochemical and biomedical properties.

**Additional Feedback:**

1. It is recommended to structure the datasets according to “task solvability” or “application scenarios”, such as beginner-level tasks vs. expert-level challenges. A simplified subset could be introduced—for example, retaining only 5–6 core fields—as an entry-level dataset for newcomers.
2. Although the data sources are from open-access literature, it is not clearly stated whether each dataset fully complies with open-source redistribution licenses (particularly regarding the reuse of tables/images from journals). It is advisable to add a section in the official documentation addressing data source licensing or including a disclaimer (e.g., specifying which data is for academic evaluation only and not for commercial use).

**Dataset Code Accessibility:**

Yes

**Dataset Code Comments:**

Yes. The authors provide detailed documentation and annotations both within the paper and through external links.

**Ethical Comments:**

This study uses publicly available scientific literature data and does not involve personal privacy, human subjects, or sensitive information. All data collection and usage comply with ethical standards.

**Ethical Considerations:**

No, there are no or only very minor ethics concerns

**Final Justification:**

I recommend an Accept (5) as the authors provided a satisfactory rebuttal that adequately addressed the initial concerns.

**Limitations Weaknesses:**

1. Although ChemX defines multiple extraction tasks, current mainstream methods (including LLMs and agent-based approaches) have not yet met the high-accuracy requirements for real-world applications. Section 5 "Experimental Results" (page 7, Figure 3) shows that all models (including GPT-4o and single-agent methods) perform significantly worse in precision/recall when extracting numerical parameters compared to textual ones. Evaluation metrics could be improved by defining an "acceptable error threshold" and incorporating task context.
2. The current ChemX experiments still rely on GPT-4o’s "black-box" capabilities for image processing (Section 4.2), rather than using controllable computer vision (CV) tools. While this is convenient, it affects both interpretability and stability. A possible improvement would be to "decouple" image processing from the language model and introduce specialized image analysis modules.
3. Missing values and incomplete fields are characteristic of real-world scenarios. As shown in Section 3.4 "Dataset Overview" (page 5, Figure 2C), some datasets have a "very high proportion of missing values" (over 40%). Section 4 uses precision, recall, and F1 scores for evaluation but does not specify how samples with partially missing fields are weighted during evaluation. Without appropriate weighting, evaluation may overestimate the performance of extraction systems (i.e., performing well on easy fields while skipping difficult ones altogether). One possible solution is to use field-weight-normalized F1 or evaluation metrics with missing-value penalties, such as weighted macro-F1. Additionally, the documentation should clarify the coverage of task-specific fields in each dataset to avoid misleading evaluations.

**Strengths Contributions:**

- All methods generally perform better at extracting textual information (e.g., names, structural formulas) than numerical data (e.g., concentrations, sizes).
- The extraction performance on small molecule datasets is better than that on nanomaterial datasets.
- Current mainstream LLMs (e.g., GPT-4o) still lack sufficient accuracy in extracting chemical information for practical applications.
- Domain-specific systems (e.g., nanoMINER) achieve good performance but are limited to specific tasks and domains.
- Ten reproducible, high-quality datasets are provided, including metadata and standardized formats, and are publicly hosted on HuggingFace and GitHub.
- A benchmarking platform has been established to promote algorithm evaluation and advancement in the field of chemical information extraction.

---

> ### Author Rebuttal · Authors · 2025-07-30
>
> We sincerely thank the reviewer for their thoughtful and detailed feedback. However, we believe the first two raised points stem from a misunderstanding of ChemX’s contributions and design. Below, we address the main points raised, and outline our planned clarifications and improvements for the camera-ready version.
>
> 1. The fact that current methods (including LLMs and agent-based approaches) struggle with high-accuracy extraction, particularly for numerical parameters – is precisely what ChemX highlights. This is not a limitation of our work (as we do not propose a new method) but rather a key strength of our dataset collection: it exposes critical challenges in real-world chemical data extraction, motivating future research. The performance gap underscores the need for improved metrics (e.g., error thresholds) and better contextual modeling, which ChemX enables by providing a rigorous benchmark.
>
> 2. The reviewer’s suggestion to decouple image processing from LLMs aligns with our single-agent design, which explicitly separates text and image analysis (Section 4.2.1). However, our results demonstrate that the core challenge lies not in system architecture but in the inherent difficulty of extraction tasks even with modular approaches. Design of our single-agent approach intentionally avoids black-box reliance on GPT-4o’s vision capabilities, and our experiments validate that the bottleneck is task complexity, not pipeline integration (Section 5, 8.3).
>
> 3. We agree with the reviewer that real-world scientific literature frequently includes incomplete information. In ChemX, we embrace this characteristic by intentionally including missing values to simulate realistic extraction challenges (see Section 3.4, Figure 2C). Importantly, our evaluation pipeline is designed to expect and handle such cases. Specifically, when a field is absent in the source document (e.g., a concentration or particle size is not reported), the correct behavior of the extraction system is to return a "NOT_DETECTED" placeholder. This convention is explicitly stated in our prompt templates and enforced during evaluation (Section 4, Prompt Design). Therefore, extraction systems are not penalized for returning "NOT_DETECTED" when values are genuinely missing in the article, but they are penalized for hallucinating incorrect values or failing to detect those that are present.
> We will make this behavior more explicit in both the paper and accompanying documentation to ensure transparency and reproducibility in how missing values are handled and scored during evaluation.
>
> Additional feedback:
>
> 1. We thank the reviewer for the excellent suggestion to organize datasets by "task solvability" or difficulty level! In the revised documentation, we will annotate each dataset with metadata indicating task complexity (e.g., “Beginner,” “Intermediate,” “Expert”) based on the number of fields, modality diversity, and extraction accuracy. Additionally, we will release a simplified starter pack consisting of 3–4 datasets with reduced field complexity to support entry-level experimentation and educational use cases.
>
> 2. We thank the reviewer for pointing out the importance of license compliance. All papers used in the creation of ChemX were either sourced from open-access journals or accessed through institutional subscriptions permitted for academic use. A dedicated section addressing data licensing and redistribution terms will be added to the documentation, clarifying compliance with open-access policies. Proper attributions and disclaimers will also be included where applicable.
>
> We hope that our responses have sufficiently addressed the reviewer's concerns. In view of the provided clarifications, we kindly ask the reviewer to update the evaluation score. Should any additional questions arise, we remain fully committed to providing further clarification or elaboration as needed.

---

### Official Review · Reviewer_uof9 · 2025-06-30

**Rating:** 4
**Confidence:** 4

**Summary:**

The paper introduces a benchmark collection for the chemical domain named ChemX, which includes 10 manually curated datasets covering the fields of small molecules and nanomaterials. These datasets have been cross-validated by domain experts and are designed to evaluate and optimize the performance of automated information extraction methods. The paper also assesses the performance of current state-of-the-art LLMs and agent methods in chemical information extraction tasks and discusses the relevant challenges and future directions.

**Dataset Code Accessibility:**

Yes

**Dataset Code Comments:**

The authors have made significant efforts to ensure the accessibility, usability, and reproducibility of their work:

1. The ChemX datasets are hosted on Hugging Face ([link](https://huggingface.co/collections/ai-chem/chemx-6820df9ecf568b1ff0ea2431)), providing structured access to all 10 datasets.Each dataset includes standardized schemas, metadata, and documentation, ensuring clarity in usage.

2. The GitHub repository ([link](https://github.com/ai-chem/ChemX)) contains:

   - Extraction pipeline code (e.g., PDF preprocessing, LLM-based extraction).
   - Evaluation scripts (precision, recall, F1-score calculations).
   - Prompt templates for different datasets (Section 8.5).

   - The marker-pdf SDK is used for structured text extraction, improving reproducibility.

**Ethical Considerations:**

No, there are no or only very minor ethics concerns

**Final Justification:**

Though my concerns regarding the experimental evaluation—for instance, whether replacing GPT-4.1-mini with domain-specific chemistry models (such as ChemDFM [1] and ChemLLM [2]) in single-agent systems could yield performance improvements—and the ChemX's current limited scope still exist.  I recognize the significant contribution of ChemX in providing ten high-quality, multimodal chemistry datasets, all of which are publicly accessible with comprehensive documentation and metadata to facilitate reproducible research. Given these merits and the potential impact on the research community, I am inclined to revise my rating to a 4.

[1] Zhao Z, Ma D, Chen L, et al. ChemDFM: a large language foundation model for chemistry[J]. arXiv preprint arXiv:2401.14818, 2024.
[2] Zhang D, Liu W, Tan Q, et al. Chemllm: A chemical large language model[J]. arXiv preprint arXiv:2402.06852, 2024.

**Limitations Weaknesses:**

1. The datasets focus on small molecules and nanomaterials, but do not cover other important areas (e.g., polymers, organometallics, biochemical pathways).

2. Data is sourced from peer-reviewed literature, which may introduce bias toward well-studied compounds and exclude negative results or proprietary data.

3. While multimodal (text, tables, figures), some key chemistry data formats (e.g., spectra, reaction SMILES, 3D molecular structures) are not included.

4. The evaluation only tested the commercial model GPT-4, without including open-source alternatives such as:

   - Domain-specific models for chemistry: ChemBERTa, MolBERT
   - General-purpose models: LLaMA-3, Mistral
   - Multimodal models

   Furthermore, no comparison was made with traditional chemical information extraction tools.

5. The workflow still requires manual validation, reducing scalability.

**Strengths Contributions:**

1. This work provides ten high-quality, multimodal chemistry datasets covering small molecules and nanomaterials domains. Each dataset has undergone rigorous expert validation and standardized processing. All datasets are publicly available with comprehensive documentation and metadata to support reproducible research.

2. It systematically evaluates the performance of LLMs (e.g., GPT-4o) and agent-based methods in chemical information extraction tasks, revealing the limitations of current technologies.

3. It establishes the first comprehensive benchmark for chemical information extraction, filling a critical gap in existing research. It discusses challenges in data quality control and extraction performance, providing valuable guidance for future studies.

---

> ### Author Rebuttal · Authors · 2025-07-30
>
> We sincerely appreciate your thoughtful review and valuable feedback on our work. Below, we provide a detailed response to your comments and concerns.
>
> 1. We acknowledge that the current version of ChemX focuses primarily on small molecules and nanomaterials. While we recognize the importance of other chemical domains (e.g., polymers, organometallic compounds, and biochemical pathways), our dataset construction relied on manual curation and expert validation, which imposes practical limitations on immediate expansion. We would also like to emphasize that this represents the first collection of such datasets, the culmination of nearly five years of dedicated effort by the research team. While future expansions into additional domains are anticipated, the current iteration constitutes a substantial contribution to the advancement of automated data extraction methodologies.
>
> 2. The absence of negative results in published research reflects a broader systemic challenge within the scientific community, where there exists a prevailing tendency to prioritize positive findings. Nevertheless, the relevance of our research framework remains important, as evidenced by numerous documented cases in which literature-derived data enabled the training of predictive models leading to novel scientific discoveries, e.g. in [1-3]. Consequently, this publication bias does not undermine the substantive contribution of our work.
>
> 3. While spectra, 3D molecular representations, and reaction schemes are valuable data types, they are beyond the scope of the current study. We will prioritize these in future updates.
>
> 4. The primary objective of this work is to establish a high-quality collection of benchmarking datasets to facilitate the evaluation and advancement of automated data extraction systems. Domain-specific models for chemistry like ChemBERT or MolBERT are primarily designed for chemical text-based tasks and lack native multimodal capabilities for processing images, graphs, or structured data. So we evaluated GPT-4o, currently recognized as a leading multimodal model, and observed that its performance on our benchmark fell significantly below expectations. Given these findings, it appears unlikely that less advanced general-purpose models would yield better results. Recognizing the limitations of general-purpose LLMs in multimodal chemical information extraction, we adopted agent-based methodologies, including evaluation of a SOTA solution from Future House that still demonstrates poor performance in processing heterogeneous chemical data formats. All these results confirm the value of ChemX as a resource for benchmarking automated extraction systems.
> We excluded traditional data extraction methods from the comparative analysis because they are inherently unimodal and not suitable for our problem formulation, where the input consists of PDF articles and the expected output is a structured list of dictionaries containing target characteristics. The primary objective of ChemX is to benchmark automated data extraction systems, a task for which traditional methods are not designed for.
>
> 5. We would like to emphasize again that our research focuses not on workflow development, but rather on the creation of a carefully curated collection of datasets designed to benchmark data extraction methodologies. These datasets were manually compiled and rigorously verified to ensure high-quality benchmarks for evaluating future generations of automated extraction systems. By providing these standardized datasets, we aim to facilitate the automated validation of such systems using ChemX, thereby supporting reproducible and scalable advancements in the field.
>
> We believe ChemX provides a rigorous, expert-validated foundation for multimodal chemical information extraction, despite its current scope limitations. Your feedback has been invaluable in shaping our roadmap for future improvements.
> Some of the expressed criticism was likely caused by a misunderstanding. In view of the clarifications provided, we kindly request reconsideration of the "borderline reject" assessment, as the benchmark’s methodological rigor, expert validation, and open accessibility represent a significant contribution to the field.
>
> References:
>
> 1. N. Shirokii, Y. Din, I. Petrov, Y. Seregin, S. Sirotenko, J. Razlivina, N. Serov, and V. Vinogradov. Quantitative prediction of inorganic nanomaterial cellular toxicity via machine learning. Small, 19(19):2207106, 2023.
>
> 2. K. A. Kapranova, J. Razlivina, A. Dmitrenko, D. V. Kladko, and V. Vinogradov. Prediction of exchange bias for magnetic heterostructure nanoparticles with machine learning. The Journal of Physical Chemistry C, 2025. Early Access.
>
> 3. J. Razlivina, A. Dmitrenko, and V. Vinogradov. Ai-powered knowledge base enables transparent prediction of nanozyme multiple catalytic activity. The Journal of Physical Chemistry Letters, 15(22):5804–5813, 2024.

---

> > ### Comment · Reviewer_uof9 · 2025-08-01
> >
> > Thank you for your feedback, and appreciate the additional details you have provided. Though my concerns regarding the experimental evaluation—for instance, whether replacing GPT-4.1-mini with domain-specific chemistry LLMs (such as ChemDFM [1] and ChemLLM [2]) in single-agent systems could yield performance improvements—and the ChemX's current limited scope still exist.  I recognize the significant contribution of ChemX in providing ten high-quality, multimodal chemistry datasets, all of which are publicly accessible with comprehensive documentation and metadata to facilitate reproducible research. Given these merits and the potential impact on the research community, I am inclined to revise my rating to a 4.
> >
> > [1] Zhao Z, Ma D, Chen L, et al. ChemDFM: a large language foundation model for chemistry[J]. arXiv preprint arXiv:2401.14818, 2024.
> > [2] Zhang D, Liu W, Tan Q, et al. Chemllm: A chemical large language model[J]. arXiv preprint arXiv:2402.06852, 2024.

---

### Official Review · Reviewer_cpxC · 2025-07-02

**Rating:** 5
**Confidence:** 4

**Summary:**

* This paper introduces ChemX, a benchmark for assessing methods that can perform automated information extraction of nanomaterial/small molecule information from scientific papers.
* ChemX consists of 10 datasets (summarized in Table 1): 5 focused on nanomaterials, 5 on small molecules. Each dataset is focused on a particular property domain. For instance, the “Eye Drops” dataset focuses on drug permeability data across corneal tissues (for small molecules).
* Dataset sizes range from 70-5535 items in size (average 2164) and are sourced from over 1500 articles.
* The authors curate the datasets in ChemX manually (see lines 127-128), with an additional cross-validation check done on around 20% of the entries (Section 3.3).
* The authors also evaluate the extraction abilities of current models on this dataset, including LLMs (GPT-4o), an “agentic” method (which converts the initial pdfs into a structured text format first), and a multi-agent approach (using nanoMiner [45]). These methods are shown to have limitations in currently being able to complete the tasks successfully.

**Additional Feedback:**

**i.** (line 210) I was a bit confused why a mini-model  was used as the final extraction method in the single-agent approach (rather than the same GPT-4o model that is used elsewhere)? Did you try using the GPT-4o method for this extraction?

**ii.** For the standard deviation reported in Table 2, is this just computed over the examples, or does this also consider the stochasticity inherent in model calls?

**iii.** I don’t understand the comment on lines 236-237 about why the small molecule datasets are expected to be easier than the nanomaterial datasets due to their lower number of “parameters”. Do you mean the number of columns/properties being extracted or are the papers also longer?

**iv.** Is there any intuition for why the methods exhibited better performance on textual parameters (line 228) compared to numeric?

**very minor nits**.
a. It would be nice to quantify “reasonably accurate output” on line 281.
b. Could write out the url of “link” on line 114.
c. Line 110: think second “X” is meant to be 10?

**Dataset Code Accessibility:**

Yes

**Dataset Code Comments:**

Data code available on GitHub (although I have not tried running this myself). This includes the code for evaluating the LLM used (and its snapshot is specified in the main paper). DOIs for the papers behind the dataset are provided, so that they can be independently downloaded. The paper also provides a website providing more information on tasks and data.

The HuggingFace datasets website includes the licences (although I was not sure if this was also for the information taken from the papers behind a paywall?).

**Ethical Comments:**

Paper is a benchmark for information extraction on published literature for chemistry-related tasks. I do not believe data from human-participants was involved. Paper aims to help evaluate capabilities for information extraction and does not claim existing models can be used without oversight.

**Ethical Considerations:**

No, there are no or only very minor ethics concerns

**Final Justification:**

I still think this paper should be accepted (and so maintain my score), due to the important problem that hopefully this benchmark will enable method development for, the range of different tasks considered, and the efforts the authors have made in ensuring data quality.

I am pleased the authors will take on board my comments about aspects of the paper that I felt could be better presented, which resolves that comment in my original review. While I am slightly disappointed that additional existing methods were not evaluated here (a point also brought up by Reviewer uof9), hopefully this benchmark allows others to follow up on this.

**Limitations Weaknesses:**

**W1 Some aspects of the paper could be better explained**
I found some of the aspects of the paper hard to follow. For instance, although figure 1 is helpful in explaining the pipeline followed, it would be nice for this to be grounded with examples from a real paper if possible. (Looking at some of the papers/extracted data, it seems some of the molecule identification tasks are particularly challenging due to references to other figures and R-Group definitions, which I did not initially appreciate from the paper alone). Likewise, it would be nice to have full details on the errors found (Figure 5 only shows two datasets), and have this information put in context with the number of examples checked.

In a related point, the aggregated performance metrics in the main paper seem to hide the fact that performance often differs widely on the different columns/properties being predicted (see Table 4). For instance, in the molecular tasks the prediction of SMILES often seems to be hard, whereas compound id identification appears much easier. This makes it hard to work out which capabilities of current methods need to be improved and whether the high variance reported actually reflects a multimodal distribution over the different task difficulties.

**W2 Limited evaluation of current models.** In the benchmark (Section 5), the authors only consider an LLM and a “single-agent” method (an exception is with the Nanozymes dataset, where nanoMINER is also compared against). Furthermore, the single agent method is more of a fixed pipeline, where the pdf is converted into a structured text format in a fixed and deterministic manner before being fed through to a final extraction model (instead of allowing the extraction model to call the tools as and when they are needed). It would be nice to see if more recent, e.g., reasoning models, might do better on the tasks. (There are some preliminary experiments on this direction in Section 6.3 using Crow [48,49])

**W3 A significant proportion of the papers making up the datasets are not open-access — limiting the ease of use of the benchmark.** Many of the papers from which the data is extracted from are not open access (exceptions being Seltox and Synergy — see Figure 2B). I appreciate that much useful data might be behind a paywall, but this unfortunately means that it may be hard for others to use the full benchmark. In fact this even limits the evaluation the authors do in Section 4, where they restrict the evaluation to open-access data only (see line 186). It would be nice if the authors could give exact numbers for the amounts of open-access examples that are actually considered in the experiments (it’s hard to parse this from Fig2B) and if they could comment on the expected variance in the figures reported due to the lower amounts of data.

**Strengths Contributions:**

**S1 Range of data with several examples for different tasks, helps with the paper's potential impact/relevance.** The different properties (both numerical and text-based) that are extracted make the benchmark an interesting challenge, testing different aspects of models. For instance, structure extraction often tests models ability to recognise and reason about images. Likewise, considering both nanomaterial and small molecule tasks (Table 1) and pdf and image inputs (Fig 3) allows one to assess models on different tasks/modalities.

**S2 Benchmark shows current methods have lots of room for improvement, helping with paper’s potential significance.** The reported Precision/Recall and F1 scores (e.g., Figure 3 or Table 2) suggest that current methods have significant room for improvement, and therefore suggests that this benchmark is useful in terms of directing the development of better models.

**S3 Data validation and quality control procedure seems sensible for ensuring information is extracted correctly.** The quality control procedure, laid out in Section 3.3 (randomly selecting ~20% of entries and ensuring they are correct), increases the likelihood that the data extracted for benchmarking is correct, and seems a sensible measure for reducing the chance that a model is penalized incorrectly (without the high costs of double checking _every_ example). (I was a bit confused about how the systematic errors came about though, given the initial manual curation?).

**S4 Benchmark seems novel compared to others that consider either inherent chemical reasoning ability in LLMs or that focus on method development for chemical data extraction.** Benchmark seems different to others curated (e.g., the task specific ones cited [35-40] or other recent benchmarks for evaluating capabilities of LLMs in chemistry, e.g., ChemBench — Mirza et al. 2024 https://arxiv.org/abs/2404.01475 ). Also different to papers which focus more on method development for different (often smaller) aspects of extraction (such as [50] or Fan et al. 2024 https://pubs.acs.org/doi/full/10.1021/acs.jcim.4c00572). Having a benchmark to help drive method development seems useful and new.

---

> ### Author Rebuttal · Authors · 2025-07-30
>
> Thank you for your thorough and constructive review of our paper. We appreciate your thoughtful feedback and will address your comments in detail below.
>
> **W1**. We appreciate your valuable feedback. In the camera-ready version, we will incorporate examples into Figure 1 and include the remaining graphs with error statistics across all datasets.
> We acknowledge that aggregated metrics may obscure certain nuances, as extraction accuracy varies significantly depending on the object type. While it is not feasible to include all metrics in the main text, we have outlined the general trends in Sections 5 and 6.2. Additionally, we have explicitly noted outlier results (Lines 228–230) to ensure transparency.
> Your comment is duly noted, and we will provide a more detailed discussion of the extraction results in the final version of the manuscript.
>
> **W2**. The primary objective of this work is to establish a high-quality benchmark to facilitate the evaluation and advancement of automated information extraction systems. We evaluated both a SOTA general-purpose LLM and a specialized SOTA multi-agent approach designed for chemistry (FutureHouse); however, neither yielded satisfactory extraction results (Section 5, 6.3). FutureHouse exhibited significant failures – requiring 16 minutes and returning mostly null (NaN) values – when processing an article with two nanozyme samples, in stark contrast to nanoMINER [1], which shows superior performance being the SOTA multi-agent system tailored to nanomaterial information extraction. However, nanoMINER lacks generalizability and cannot be readily adapted to other related scientific fields – a key limitation that our work underscores. Given the limited performance observed with these approaches, we did not pursue additional methods in this study.
> Overall, our findings clearly highlight the critical need for a foundational resource to drive future improvements in information extraction methodologies. To address it, we introduced ChemX for the small molecule and nanomaterial domains.
>
> **W3**. We appreciate the reviewer’s valid point regarding the accessibility of papers in our benchmark. Below, we provide a detailed breakdown of open-access (OA)  papers across datasets:
>
> - Benzimidazole: 9 OA
> - Cocrystals: 13 OA
> - Complexes: 3 OA
> - Cytotoxicity: 58 OA
> - Magnetic: 120 OA
> - Nanozymes: 39 OA
> - Oxazolidinone: 2 OA
> - Seltox: 135 OA
> - Synergy: 54 OA
>
> The predominance of non-OA papers reflects the reality of chemical literature (much high-value data resides in paywalled journals). While OA subsets are smaller, they are distributed across all property types (e.g., SMILES, numerical values) and thus reflect the full benchmark’s diversity. Furthermore, we provide experimental results using OA articles to ensure full reproducibility of our findings. While this study focuses on OA publications, many academic institutions maintain subscriptions to licensed journals, enabling researchers to utilize the complete version of ChemX for benchmarking automated data extraction methods across a broader range of scientific literature.
>
> **Additional Feedback**:
>
> i. Our comparative evaluation revealed that GPT-4.1-mini demonstrated superior performance to GPT-4o in processing scientific chemical texts, particularly for named entity recognition (NER) and structured data extraction tasks. This advantage could only be used in a single agent approach, which first converts the input PDF to text and then performs the extraction. When tested on manually-validated samples from research articles, GPT-4.1-mini achieved higher accuracy in both chemical entity identification and structural information extraction, suggesting its greater suitability for domain-specific scientific text processing compared to the more general GPT-4o implementation. For example, in the benzimidazole dataset, the GPT-4o model does not find the unit of measurement of the target parameter.
>
>   **GPT-4o**
>
>     "compound_id": "1",
>
>     "smiles": "c1ccc2c(c1)ncc2/C=C/C3=N/CCN3",
>
>     "target_type": "pMIC",
>
>     "target_relation": "=",
>
>     "target_value": 0.73,
>
>     "target_units": "NOT_DETECTED",
>
>     "bacteria": "Escherichia coli"
>
>  **GPT-4.1-mini**
>
>     "compound_id": "1",
>
>     "smiles": "c1ccc2c(c1)ncc2/C=C/C3=N/CCN3",
>
>     "target_type": "pMIC",
>
>     "target_relation": "=",
>
>     "target_value": 0.73,
>
>     "target_units": "µM",
>
>     "bacteria": "Escherichia Coli"
>
> Similar errors are observed in other datasets. We will provide more examples in the camera-ready version of the article.
>
> ii. The standard deviation in Table 2 reflects variability across examples only, not model stochasticity. We will clarify this in the text.
>
> iii. The "parameters" refer to the number of unique properties (columns) being extracted. Small molecule datasets generally exhibit fewer interdependent features within the text of scientific papers, thereby reducing the likelihood of model misclassification or feature confusion.
>
> iv. The observed disparity in extraction performance between numerical and textual data likely stems from inherent challenges in processing numeric values, particularly due to OCR errors in tables and figures. Numerical data embedded in tabulated formats or images is highly susceptible to misrecognition during PDF parsing, which can alter digit sequences (e.g., "0.45" being misinterpreted as "0.65").
>
> **Minor points**:
>
> a. We will quantify "reasonably accurate output" (line 281) with a concrete example in the camera-ready version.
>
> b. The URL (line 114) will be written in full.
>
> c. Yes,  "X" (line 110) should indeed be the Roman numeral "X" representing the value 10.
>
> We sincerely appreciate the reviewer's careful reading of our manuscript, valuable comments, and the positive assessment of our work!
>
> References:
>
> 1. Odobesku, R., Romanova, K., Mirzaeva, S. et al. Agent-based multimodal information extraction for nanomaterials. npj Comput Mater 11, 194 (2025). https://doi.org/10.1038/s41524-025-01674-7

---

> > ### Comment · Reviewer_cpxC · 2025-08-01
> > **Response to rebuttal**
> >
> > Thanks a lot to the authors for their rebuttal and response to my questions!
> >
> > I think the authors did a good job in responding to the weaknesses I listed in the original paper:
> > **W1.** Mainly about the presentation both in terms of how the data was extracted but also how the aggregate results presented hide more subtle differences in performance across the different properties (think a similar point also made by Reviewer ozaP in their point 3, but focussing more on missing values for the different properties or tasks). _=> Pleased to see that the authors will take on board comments for the final camera-ready version, and so happy to consider point resolved. (I also appreciate space constraints wrt reporting broken down vs aggregate results)._
> > **W2.** About limited evaluation of current models. _=> Authors pointed out that the current agent-based systems did not work that well or were too specialized so couldn't be included across all experiments. Whether more recent reasoning models work better is left open, but hopefully the benchmark will be useful in enabling follow up work to assess this._
> > **W3.** About much of the benchmark being behind a paywall. _=> Appreciate the values on the numbers of open access papers considered (433 total), which helps put the experiments in context._
> >
> > Also, thanks for answering the questions I had in the additional feedback section. Just had a couple of follow-up questions:
> > 1. How did the systematic errors come about? Do you have an example of one? (Still a bit confused about how the errors were linked due to the manual curation).
> > 2. Regarding point ii: Variance is across examples not model stochasticity. Thanks for clarifying. In your experience do you think model stochasticity would affect the reported results significantly, or are the models fairly consistent in their responses to the tasks posed?
> >
> > Will follow the discussion with the rest of the reviewers and will update my review.

---

> > ### Author Response · Authors · 2025-08-04
> > **Rebuttal**
> >
> > Thanks a lot for valuable comments. We will carefully consider all feedback and incorporate relevant suggestions into the final version of the manuscript.
> >
> > 1. Errors were identified manually and encompassed a range of issues, such as transcription mistakes, structural mismatches, unit inconsistencies, or values inferred by curators that were not explicitly reported in the original source. Rather than classifying errors into predefined categories, we focused on whether an error was general (meaning it followed a consistent pattern across multiple entries) or appeared to be isolated (a single instance without recurrence). Systematic errors may arise, for example, when numerical values are presented in a tabular format but the author misinterprets the units of a measurement, leading to inaccuracies across multiple samples in the dataset (E.g., "nM" misinterpreted and initially logged as "mM" by a human, corrected to "nM" during the cross-verification).
> >
> > 2. We acknowledge that the stochastic nature of the model can influence the final extraction metrics, particularly as the temperature parameter increases. Based on our observations, in tasks involving well-defined structures, such as classification, extraction, and named entity recognition, the model's output is typically unambiguous. For instance, as demonstrated by Odobesku et al. [1] (section Methods, System Evaluation), when the temperature is fixed at 0 across 100 independent extraction runs, only negligible variations in outputs are observed (Table S3).
> > We think that the contribution of the model's stochasticity to the overall standard deviation will be considerably less than the variability observed across all extracted metrics. Therefore, we have chosen to disregard the model's stochasticity in our estimation of the standard deviation.
> >
> > We appreciate your comments and questions and hope our responses have addressed them fully.
> >
> > Links:
> >
> > 1. Odobesku, R., Romanova, K., Mirzaeva, S. et al. Agent-based multimodal information extraction for nanomaterials. npj Comput Mater 11, 194 (2025). https://doi.org/10.1038/s41524-025-01674-7

---

> > > ### Comment · Reviewer_cpxC · 2025-08-06
> > > **Thank you for your reply**
> > >
> > > Thank you for answering my other questions and appreciate the clarification on the systematic errors & ignoring of model stochasticity.

---

### Decision · Program_Chairs · 2025-09-18

**Decision:**

Accept (poster)

**Comment:**

This paper introduces ChemX, a novel and well-executed benchmark suite comprising 10 manually curated datasets targeting multimodal information extraction from chemical literature. The dataset spans two crucial domains—nanomaterials and small molecules—and is derived from over 1,500 peer-reviewed scientific articles. The authors conduct a thorough evaluation of multiple extraction pipelines, including strong LLM baselines and multi-agent systems, and present valuable insights into current limitations in automated chemical information extraction. The reviewers unanimously recognized the technical quality, novelty, and potential impact of the benchmark, even as they noted some areas for improvement in evaluation breadth and presentation. This is a technically solid, well-executed paper with high utility to the AI and chemistry communities. ChemX is poised to become a foundational benchmark for evaluating multimodal information extraction methods in scientific domains. I strongly recommend acceptance.